# Domain-dependent effects of insulin and IGF-1 receptors on signalling and gene expression

Weikang Cai[1,2,*], Masaji Sakaguchi[1,2,*], Andre Kleinridders[1,2,3,4], Gonzalo Gonzalez-Del Pino[5,6], Jonathan M. Dreyfuss[7,8], Brian T. O'Neill[1,2,9], Alfred K. Ramirez[1,2,8], Hui Pan[7,8], Jonathon N. Winnay[1,2], Jeremie Boucher[1,2,10], Michael J. Eck[5,6] & C. Ronald Kahn[1,2]

Despite a high degree of homology, insulin receptor (IR) and IGF-1 receptor (IGF1R) mediate distinct cellular and physiological functions. Here, we demonstrate how domain differences between IR and IGF1R contribute to the distinct functions of these receptors using chimeric and site-mutated receptors. Receptors with the intracellular domain of IGF1R show increased activation of Shc and Gab-1 and more potent regulation of genes involved in proliferation, corresponding to their higher mitogenic activity. Conversely, receptors with the intracellular domain of IR display higher IRS-1 phosphorylation, stronger regulation of genes in metabolic pathways and more dramatic glycolytic responses to hormonal stimulation. Strikingly, replacement of leucine[973] in the juxtamembrane region of IR to phenylalanine, which is present in IGF1R, mimics many of these signalling and gene expression responses. Overall, we show that the distinct activities of the closely related IR and IGF1R are mediated by their intracellular juxtamembrane region and substrate binding to this region.

[1] Section of Integrative Physiology and Metabolism, Joslin Diabetes Center, Boston, Massachusetts 02215, USA. [2] Department of Medicine, Harvard Medical School, Boston, Massachusetts 02215, USA. [3] German Institute of Human Nutrition, Central Regulation of Metabolism, Potsdam-Rehbrücke, 14558 Nuthetal, Germany. [4] National Center for Diabetes Research (DZD), 85764 Neuherberg, Germany. [5] Department of Cancer Biology, Dana-Farber Cancer Institute, Boston, Massachusetts 02215, USA. [6] Department of Biological Chemistry and Molecular Pharmacology, Harvard Medical School, Boston, Massachusetts 02215, USA. [7] Bioinformatics Core, Joslin Diabetes Center, Boston, Massachusetts 02215, USA. [8] Department of Biomedical Engineering, Boston University, Boston, Massachusetts 02215, USA. [9] Department of Endocrinology and Metabolism, Fraternal Order of Eagles Diabetes Research Center, University of Iowa Carver College of Medicine, Iowa City, Iowa 52242, USA. [10] IMED Cardiovascular and Metabolic Diseases, AstraZeneca R&D, 43183 Mölndal, Sweden. * These authors contributed equally to this work. Correspondence and requests for materials should be addressed to C.R.K. (email: c.ronald.kahn@joslin.harvard.edu).

Insulin and insulin-like growth factor-1 (IGF-1) signalling pathways are closely related and well conserved throughout evolution. Insulin binds with high affinity to the insulin receptor (IR)[1], while IGF-1 binds with highest affinity to the IGF-1 receptor (IGF1R)[2]. For both receptors, ligand binding activates intrinsic receptor tyrosine kinase activity and downstream signalling cascades, which in turn regulate gene transcription, glucose, lipid and protein metabolism as well as cell growth and differentiation[3]. Despite the high degree of receptor homology and the use of almost identical intracellular pathways, the biological processes regulated by these two signalling pathways are strikingly distinct. Mutations in the IR in humans[4] or knockout of the IR in mice[5,6] result in severe hyperglycaemia, usually without major growth defects, whereas humans with mutations in the IGF1R[7] or mice lacking IGF1R[8] show severe growth retardation, but no major disturbance in glucose homoeostasis. These results, along with studies in vitro[9–11], demonstrate that IR primarily mediates metabolic effects, while IGF1R is more involved in the mitogenic control.

Understanding the exact molecular mechanisms responsible for the functional differences between IR and IGF1R is quite challenging. Overall, IR and IGF1R share more than 50% sequence homology and over 80% homology in the intracellular kinase domain. Both insulin and IGF-1 are able to bind to and activate each other's receptors, albeit with reduced affinity, and both IR and IGF1R also elicit common downstream signalling with phosphorylation of a family of IR substrates and activation of two major downstream signalling pathways, the phosphoinositide 3-kinase (PI3K)-Akt pathway and the Shc-Ras-MAPK pathway[3]. Separating the actions of these two hormones is even more difficult since one α/β heterodimer of the IR can disulfide bond to one α/β heterodimer of the IGF1R to form hybrid IR/IGF1R receptors[12], which can also bind both insulin and IGF-1 and elicit signalling events[13].

Many studies have tried to dissect the factors contributing to the differences between IR and IGF1R action. While some of the difference may be the result of different levels of relative expression of IR and IGF1R in different tissues or cell types, studies also suggest that the receptors themselves may display different functions. For instance, some studies have suggested differences in preference of IR and IGF1R for different insulin receptor substrates (IRS)[14], and others have suggested that insulin and IGF-1 are able to induce overlapping, but distinct, patterns of gene expression[15–17]. Studies using chimeric receptors have suggested that the IGF1R intracellular domain (IGF1R-ICD) may be more important for mitogenic response, while the IR-ICD is more strongly coupled to glycogen synthesis[9,10,18]. However, in most of these systems, the cells also express endogenous IR and IGF1R, which can form hybrid receptors with the different types of exogenously expressed receptors or signal directly, complicating the interpretation of these data.

In the present study, we attempt to elucidate the basis for differences in insulin and IGF-1 signalling by defining intrinsic domain-dependent effects of IR and IGF1R in cells expressing only IR, IGF1R or receptors in which intracellular and extracellular domains (ECDs) of these receptors had been swapped or mutated. We find that both ECDs and ICDs contribute to the differential signalling and gene transcription regulation of IR and IGF1R. The ICD of the IGF1R couples more strongly to Shc and Gab-1 activation and genes involved in cell proliferation, whereas the ICD of the IR is more potent in regulating IRS-1 phosphorylation and genes involved in metabolic pathways. By point-mutational and structural modelling, we have identified one amino acid in the juxtamembrane region of the ICD of the two receptors that determines substrate preference and plays a major role in differentiating IR and IGF1R action with respect to both signalling and gene expression.

## Results

**Regulation of mitogenesis and glycolysis by IR and IGF1R.** To dissect the mechanisms underlying their distinct functions of IR and IGF1R, we generated brown preadipocytes in which both endogenous IR and IGF1R had been genetically inactivated using Cre-lox recombination[19,20]. These IR and IGF1R DKO cells were then reconstituted with wild-type mouse IR, IGF1R or one of two chimeric receptors: IR/IGF1R with the IR-ECD fused to the IGF1R transmembrane and ICD and IGF1R/IR with the ECD of IGF1R fused to the transmembrane and ICD of IR (Fig. 1a). Three independent clones for each line were used for the study.

While DKO cells showed no detectable levels of mRNA for either IR or IGF1R, cells expressing normal IR and chimeric receptor IR/IGF1R had 14- and 10-fold higher receptor expression than endogenous IR expression in wild-type brown preadipocytes. Similarly, cells expressing IGF1R and IGF1R/IR had 17- and 12-fold overexpression of recombinant receptor expression than endogenous IGF1R expression in wild-type cells (Fig. 1b). Insulin stimulation of cells expressing IR or IR/IGF1R and IGF-1 stimulation of cells expressing IGF1R and IGF1R/IR led to robust and similar levels of receptor autophosphorylation consistent with the similar levels of overexpression (Fig. 1c).

Under normal culture conditions with 10% fetal bovine serum, DKO cells displayed a significantly slower rate of proliferation than wild-type preadipocytes (doubling time = 13.2 versus 9.5 h, $P < 0.001$). Re-expression of wild-type IGF1R almost completely rescued the mitogenic deficit in DKO cells, whereas re-expression of wild-type IR only partially, but significantly, rescued the mitogenic defect (Fig. 1d). Expression of the IR/IGF1R chimeric receptor led to a trend toward increased proliferation compared to IR, whereas the proliferation rate of IGF1R/IR-expressing cells was similar to cells expressing wild-type IR (Fig. 1d). Thus, both IR and IGF1R can significantly support serum-stimulated proliferation on brown preadipocytes. However, receptors with IGF1R-ICD (that is, IGF1R and IR/IGF1R) showed higher levels of proliferative potential compared to receptors with IR-ICD (that is, IR and IGF1R/IR) (Fig. 1e).

The opposite was true for the metabolic activity. Cells expressing normal IR showed increased insulin-induced glycolysis (1.6-fold) and maximal glycolytic capacity (2.9-fold) as assessed by extracellular acidification rate (ECAR) compared to cells expressing IR/IGF1R (Fig. 1f,g). Likewise cells expressing IGF1R/IR showed a strong trend toward increase in IGF-1-induced glycolysis rate (1.6-fold) and maximal glycolytic capacity (1.5-fold) compared to cells expressing IGF1R (Fig. 1f,g). Thus, DKO preadipocytes expressing receptors with IR-ICD showed higher metabolic activities as assessed by glycolytic rate, whereas preadipocytes expressing receptors with IGF1R-ICD displayed a stronger growth potential.

**Differential signalling by normal and chimeric receptors.** Both IR and IGF1R undergo internalization upon ligand stimulation[21]. However, insulin stimulated a more rapid receptor internalization of IR and IR/IGF1R with a ~50% reduction of the surface-labelled IR and IR/IGF1R receptors by 30 min, whereas IGF-1 led to internalization of only ~20% the surface IGF1R and IGF1R/IR by 30 min (Supplementary Fig. 1). These early differences disappeared over time, such that after 120 min stimulation, all four receptors showed similar levels of internalization (40–50%) (Supplementary Fig. 1). Thus, receptors with IR-ECD have more rapid internalization rate upon ligand binding than those with IGF1R-ECD.

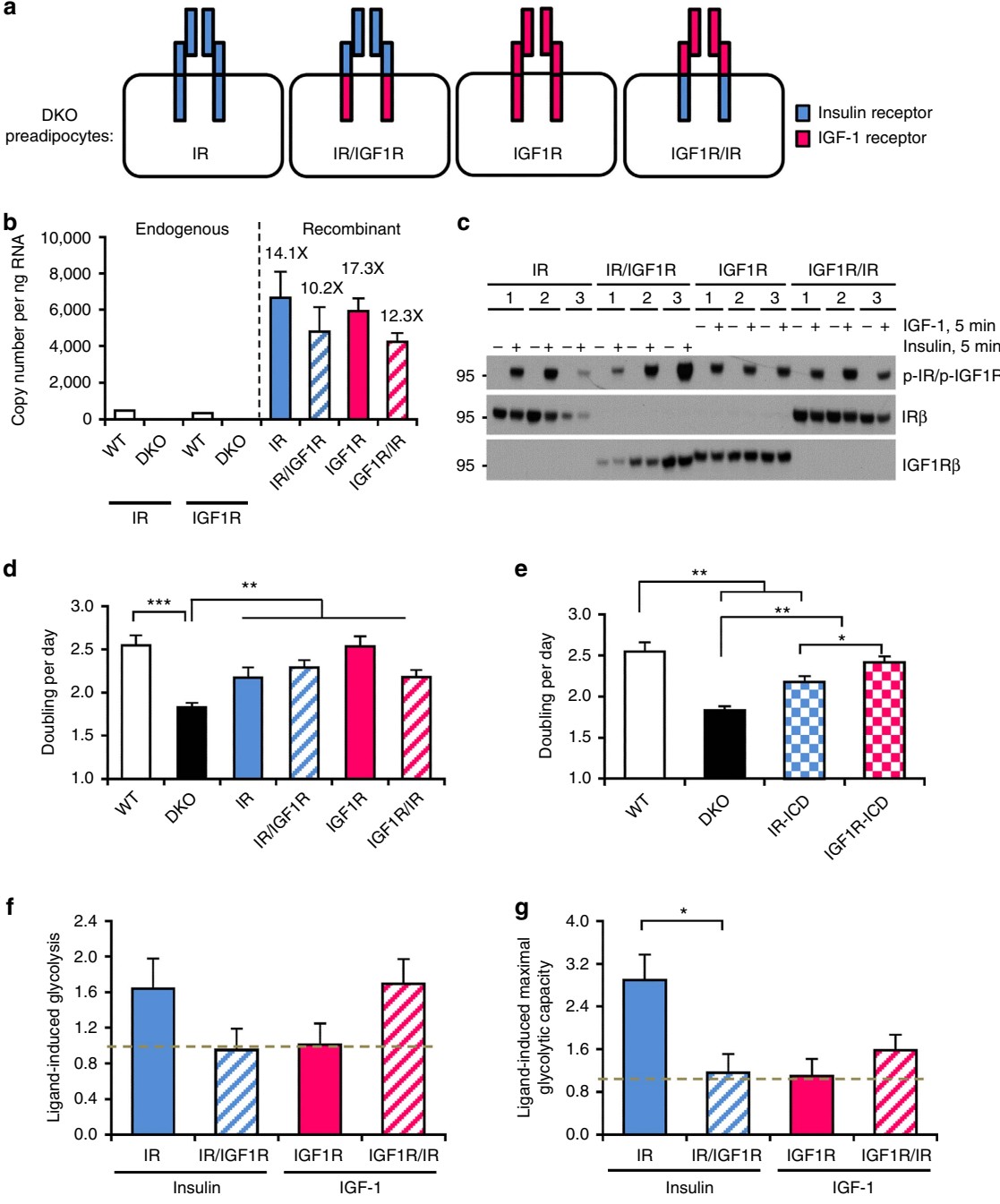

**Figure 1 | Differential roles of IR and IGF1R in regulating proliferation and glycolysis.** (**a**) Schematic showing IR/IGF1R double knockout preadipocytes reconstituted with normal IR, IGF1R, and chimeric receptors IR/IGF1R and IGF1R/IR. (**b**) Relative mRNA levels of endogenous IR and IGF1R from wild-type and DKO cells, as well as overexpressed recombinant receptors. Data were shown as mean ± s.e.m. copy number per ng total RNA, $n = 3$. Fold overexpression of each recombinant receptor to endogenous receptor are highlighted. (**c**) Immunoblotting of phosphorylated and total receptors in lysates from three independent lines of cells expressing normal IR and chimeric receptor IR/IGF1R stimulated with 10 nM insulin or cells expressing normal IGF1R and chimeric receptor IGF1R/IR stimulated with 10 nM IGF-1 for 5 min. (**d**) Proliferation rates of wild-type, DKO and receptor-reconstituted cells grown in DMEM + 10% FBS. Cell doubling times per day are shown as mean ± s.e.m. (**$P < 0.01$; ***$P < 0.001$. One-way ANOVA followed by Newman–Keuls *post-hoc* analysis, $n = 6$). (**e**) Proliferation rates of wild-type, DKO cells and combined data on cells expressing receptors with IR-ICD and IGF1R-ICD grown in DMEM + 10% FBS. Data are shown as mean ± s.e.m. (*$P < 0.05$; **$P < 0.01$. One-way ANOVA followed by Newman–Keuls *post-hoc* analysis, $n = 12$). (**f**) Glycolysis induced by insulin or IGF-1 (100 nM) stimulation for 6 h. Cells were serum starved overnight, stimulated with ligands and ECAR values as an indicator for glycolysis were measured using a Seahorse X24 Bioanalyzer. Fold change of glycolysis rates in response to stimulation were calculated for each cell line and presented as mean ± s.e.m. (Student *t*-test, $n = 5$). (**g**) Maximal glycolytic capacity induced by insulin or IGF-1 (100 nM) stimulation for 6 h. Fold change of the maximal glycolytic capacity as measured by ECAR upon ligand stimulation was calculated for each cell line and shown as mean ± s.e.m. (*$P < 0.05$, Student *t*-test, $n = 5$).

Ligand stimulation induced robust autophosphorylation of all four receptors. This was detectable as early as 5 min following stimulation for all four receptors and remained stably phosphorylated for 60 min (Supplementary Fig. 2a). Phosphorylation of downstream molecules, however, differed in their kinetics. IRS-1 and Gab-1 showed rapid tyrosine phosphorylation at 5 min, which were sustained for 60 min (Supplementary Fig. 2b,c). ERK1/2 phosphorylation displayed a biphasic kinetics showing a peak at 5 min, followed by a decline and a weaker second peak at 30 min in IR and IR/IGF1R-expressing cells (Supplementary Fig. 2d, blue lines). Furthermore, in cells expressing receptors with IGF1R-ECD (that is, IGF1R and IGF1R/IR), ERK1/2 showed more prolonged phosphorylation over the 60 min time course than cells with IR-ICD (Supplementary Fig. 2d, red lines). For all receptors, Shc, Akt and p70S6K1-S6 pathways displayed a much slower but sustained activation with a peak phosphorylation around or after 15 min of stimulation (Supplementary Fig. 2e–h). Considering the different activation kinetics, for comparison of all four receptors, we focused on the 5 min time point for the phosphorylation of IRS-1, Gab-1 and ERK1/2, and the 15 min time point for Shc, Akt, p70S6K1 and S6 phosphorylation.

While all the receptors showed robust activation, at the peak time, the amplitudes of the downstream targets phosphorylation in these four cell types were quite different (Fig. 2a,e). Phosphorylation of IRS-1 Tyr$^{612}$ was 1.9-fold higher in IR-expressing cells than that in cells expressing IR/IGF1R, whereas the phosphorylation of IRS-1 was comparable between cells expressing normal IGF1R and the IGF1R/IR chimera (Fig. 2b). Receptors with IGF1R-ICD, especially wild-type IGF1R, supported higher stimulated phosphorylation of Gab-1 and ERK1/2 (Fig. 2c,d). Likewise, Shc phosphorylation was ~3-fold higher in cells expressing receptors with IGF1R-ICD (Fig. 2f), which was paralleled by increased phosphorylation of p70S6K1 and S6 protein (Fig. 2h,i). Akt phosphorylation on Ser$^{473}$, on the other hand, was similar among all four lines (Fig. 2g). Thus, the ICD of IR was more potent in phosphorylation of IRS-1, whereas the ICD of the IGF1R played a more dominant role in activating Shc, Gab-1 and p70S6K1 signalling pathways.

**Gene expression regulation by IR and IGF1R.** To test whether the differential early signalling events of IR and IGF1R would result in differential gene expression, we serum-starved pre-adipocytes expressing wild-type IR and IGF1R overnight and stimulated these cells with 100 nM insulin, IGF-1 or vehicle for 6 h, and subjected the cellular RNA to analysis using Affymetrix Mouse Gene 2.0 ST arrays. PCA analysis showed that gene expression of the cells expressing IR and IGF1R were already distinct in the non-stimulated state and further segregated upon ligand stimulation (Supplementary Fig. 3). Of the over 30,000 genes represented on the chip, 1,228 genes were regulated by both IR and IGF1R with 724 upregulated and 504 downregulated by both receptors (FDR < 0.05 in both groups, Fig. 3a and Supplementary Fig. 4). Beyond this, 133 genes were specifically upregulated and 73 were specifically downregulated by at least 50% in IR, but not in IGF1R, expressing cells, while 180 genes were uniquely upregulated and 211 downregulated by at least 50% in IGF1R-expressing cells (Fig. 3a).

The most differentially regulated genes by IR versus IGF1R (Fig. 3b and Supplementary Table 1) included transcription factors, signalling molecules, metabolic genes and some microRNAs, demonstrating the broad control of gene transcription elicited by insulin and IGF-1. A heat map of the top 50 regulated genes at the 6 h time point is shown in Fig. 3c. These genes fell into four distinct clusters. Thirty genes, designated Group I, were specifically upregulated after stimulation of IGF1R-expressing cells with less significant changes in IR-expressing cells. These

were exemplified by heparin-binding EGF-like growth factor (Hbegf) and ERBB receptor feedback inhibitor 1 (Errfi). Group II included five genes: family with Ras association (RalGDS/AF-6) domain family member 2 (Rassf2), Gm15716, protein chibby homolog 1 (Cby1), sequence similarity 43, member A (Fam43a) and sorting nexin 29 (Snx29). They were specifically suppressed in IGF1R, but not IR, expressing cells in response to stimulation. The remainder of genes were highly regulated only in IR-expressing cells and included 13 upregulated genes (Group III), such as glycogen Synthase 1 (Gys1), phosphofructokinase, liver type (Pfkl) and Bcl2/adenovirus E1B 19 kDa interacting protein 3 (Bnip3), and two downregulated genes at 6 h Egr1 and Egr2 (Group IV). In general, IGF1R contributed more to regulation of genes involved in cell proliferation (that is, Hbegf, collagen and calcium binding EGF domains 1 (Ccbe1), Errfi1, Klf5, Cby1 and Rassf2), whereas IR more potently regulated genes involved in metabolism like Gys1, Pfkl, pyruvate dehydrogenase kinase 1 (Pdk1) and phosphoglucomutase 2 (pgm2). This was further supported by gene set analysis which indicated IR most potently regulated of metabolic pathways (glucose, pyruvate and cholesterol metabolism), as well as genes in the hypoxia inducible factor and protein turnover pathways, and genes implicated in diabetes (Supplementary Table 2). In contrast, IGF1R had a greater effect on pathways regulating proliferation, cytoskeleton dynamics, cell surface proteins and cell–cell interactions.

**Gene regulation by ECDs of IR and IGF1R.** To determine the contribution of the extracellular versus intracellular domains of the insulin and IGF-1 receptors in gene regulation, the data were further analysed by grouping cells expressing receptors with the IR-ECD (IR and IR/IGF1R) versus cells expressing receptors with IGF1R-ECD (IGF1R and IGF1R/IR) (Fig. 4a). While most genes show coordinate regulation, that is, fall along the diagonal in Fig. 4a, 141 genes (208 probe sets) were significantly (P values < 0.05) differentially regulated by at least 50% (green dots showing in Fig. 4a). This accounted for 15% of the genes uniquely regulated by IR and ~1% of the genes uniquely regulated by IGF1R (Supplementary Fig. 5). The top 20 genes preferentially regulated by the IR-ECD versus IGF1R-ECD (Group I and II) are shown in Fig. 4b and Supplementary Table 3. The responses of some representative genes were further confirmed by qPCR (Fig. 4c,d). Interestingly, the majority of the top 40 significant genes were either specifically upregulated or suppressed by IR-ECD, including cholecystokinin (Cck), Cd14 and several non-coding RNAs. Many pathways identified as specifically regulated by normal IR were also enriched in IR-ECD regulated group (Supplementary Table 4). These included pyruvate metabolism, cholesterol metabolism and protein turnover pathways. In contrast, receptors with IGF1R-ECD preferentially regulated pathways such as voltage gated potassium channels and keratin sulfate degradation. Thus, a portion of the IR-specific and IGF1R-specific effects on gene transcription appears to be dependent on the ECDs of these receptors.

**Gene regulation by ICDs of IR and IGF1R.** To identify gene expression uniquely dependent upon ICDs of the IR and IGF1R we compared cells expressing receptors with the IR-ICD (IR and IGF1R/IR) and cells expressing receptors with IGF1R-ICD (IGF1R and IR/IGF1R). By microarray analysis of ligand-stimulated cells, most genes showed coordinate regulation, however, 347 genes (366 probe sets) were significantly (P < 0.05) differentially regulated by at least 50% between cells expressing ICDs of IR and IGF1R (green points outside of dashed lines, Fig. 5a), accounting for 11% of the genes uniquely regulated by

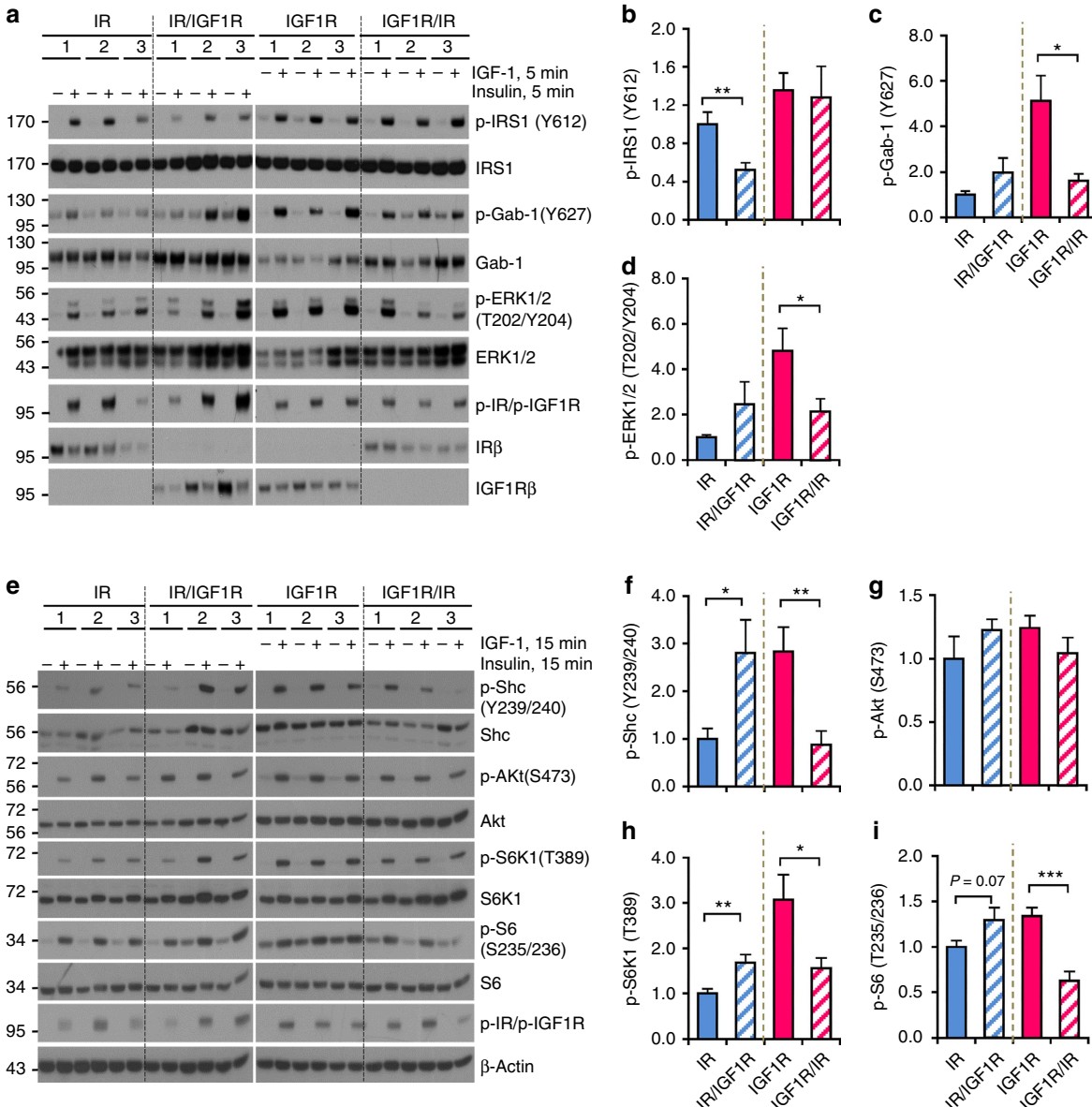

**Figure 2 | IR and IGF1R modulate different signalling upon hormonal stimulation.** (**a**) Immunoblotting of protein phosphorylation in lysates from cells stimulated with insulin or IGF-1 (10 nM) for 5 min. (**b**–**d**) Densitometric analysis of phosphorylated proteins following 5 min after stimulation. Data are mean ± s.e.m. (*$P < 0.05$; **$P < 0.01$. One-way ANOVA followed by Newman–Keuls *post-hoc* analysis; three clones used in three independent experiments). (**e**) Immunoblotting of protein phosphorylation in lysates from cells stimulated with insulin or IGF-1 (10 nM) for 15 min. (**f**–**i**) Densitometry analysis of phosphorylated proteins following 15 min ligand stimulation. Data are mean ± s.e.m. (*$P < 0.05$; **$P < 0.01$. One-way ANOVA followed by Newman–Keuls *post-hoc* analysis; three clones used in three independent experiments).

IR and 31% of the genes uniquely regulated by IGF1R (Supplementary Fig. 5). Fifty of the most differentially regulated genes are shown as a heat map in Fig. 5b (see details in Supplementary Table 5), and the differential stimulation/ suppression of several of these genes was confirmed by qPCR (Fig. 5c–f). Cells expressing receptors with IGF1R-ICD had more potent effects on gene expression after ligand stimulation for 43 of the 50 most regulated genes, enhancing expression of 28 genes and suppressing 15 (Fig. 5b, Group I and II). Genes involved in proliferative processes, such as *Hbegf* and *Ccnd1*, were stimulated in an IGF1R-ICD-dependent fashion (Fig. 5c), while anti-proliferative genes, like *Cdkn2d* and *Rassf2*, were suppressed (Fig. 5d), consistent with the higher mitogenic activity of cells expressing receptors with an IGF1R-ICD (Fig. 1e). In contrast, of these most regulated genes, only five genes were induced

in cells expressing receptors with IR-ICD, including *S1pr1*, *Serpinb1a* and *Pfkl* (Fig. 5b Group III and Fig. 5e). Interestingly, the early response genes *Egr1* and *Egr2* were more highly induced in cells expressing receptors with IGF1R-ICD at the early time point (30 min) (Supplementary Fig. 6), but dramatically sup-pressed in the cells expressing receptors with IR-ICD 6 h after stimulation (Fig. 5b Group IV, Fig. 5f and Supplementary Fig. 6), indicating differential and bidirectional regulation of *Egr1* and *Egr2* controlled by the ICDs of IR and IGF1R.

Gene set analysis further supported the differential gene expression patterns between cells expressing receptors with IR-ICD versus IGF1R-ICD. Thus, similar to the difference with the native receptors, cells expressing receptors with IR-ICD showed more control over genes involved in metabolic pathways, such as glucose metabolism (Supplementary Table 6), whereas

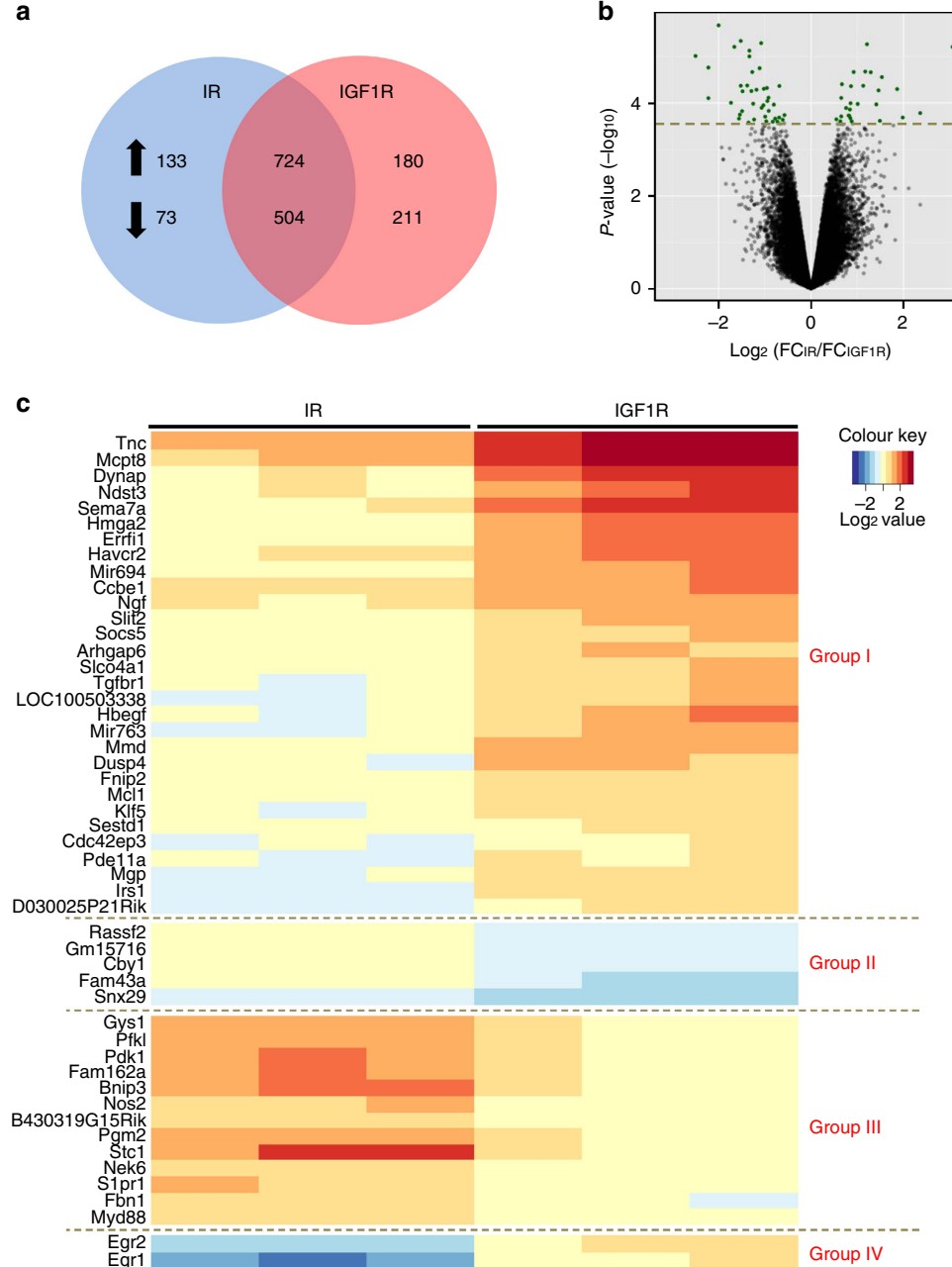

**Figure 3 | Differential gene expression patterns regulated by IR and IGF1R. (a)** Venn diagram showing the numbers of significantly regulated genes in IR and IGF1R-expressing cells. Left part of the IR pie represents the genes specifically regulated by IR by at least 50% (FDR < 0.05). Right part of the IGF1R pie represents the genes specifically regulated by IGF1R by at least 50% (FDR < 0.05). The overlapping region represents genes regulated by both IR and IGF1R (FDR < 0.05). **(b)** Volcano plot showing the distribution of differentially regulated probe sets in IR and IGF1R, with log ratio of fold change in IR versus fold change in IGF1R on x axis and $-\log_{10} P$ value on y axis. Each dot represents the mean of all the clones for each type of receptor. Highlighted dots represent the most significant genes. **(c)** Heat map of top 50 significant genes differentially regulated by IR and IGF1R.

cells expressing receptors with IGF1R-ICD were more potent in the regulation of genes involved in control of cell cycle, cytoskeletal dynamics, glycoprotein synthesis and mitochondrial fatty acid beta oxidation.

**Differences in the juxtamembrane regions on signalling.** The NPEY motif within the juxtamembrane region of the IR has previously been shown to be critical for the recruitment of receptor substrates, including Shc and IRS-1 (refs 22,23). However, in the region surrounding the NPEY motif, 4 out of 16 residues differ between the IR and IGF1R (Fig. 6a). With the exception of serine at Y + 4 position in the chicken IGF1R, these

amino acid differences between IR and IGF1R are evolutionarily conserved from chicken to humans. We hypothesized that these residue differences from IR and IGF1R might be key to the differential recruitment of IRS-1 and Shc, and thereby the activation of different downstream signalling and gene expression pathways.

To test this hypothesis, we substituted these four variable residues in IR-B isoform to the corresponding residues in IGF1R and compared their IRS-1 and Shc recruitment with wild-type IR. All of the mutated IRs showed normal phosphorylation at the tyrosine residues in the kinase domain following insulin stimulation (Fig. 6b). Both IRS-1 and Shc co-immunoprecipitated with wild-type IR and showed robust phosphorylation in

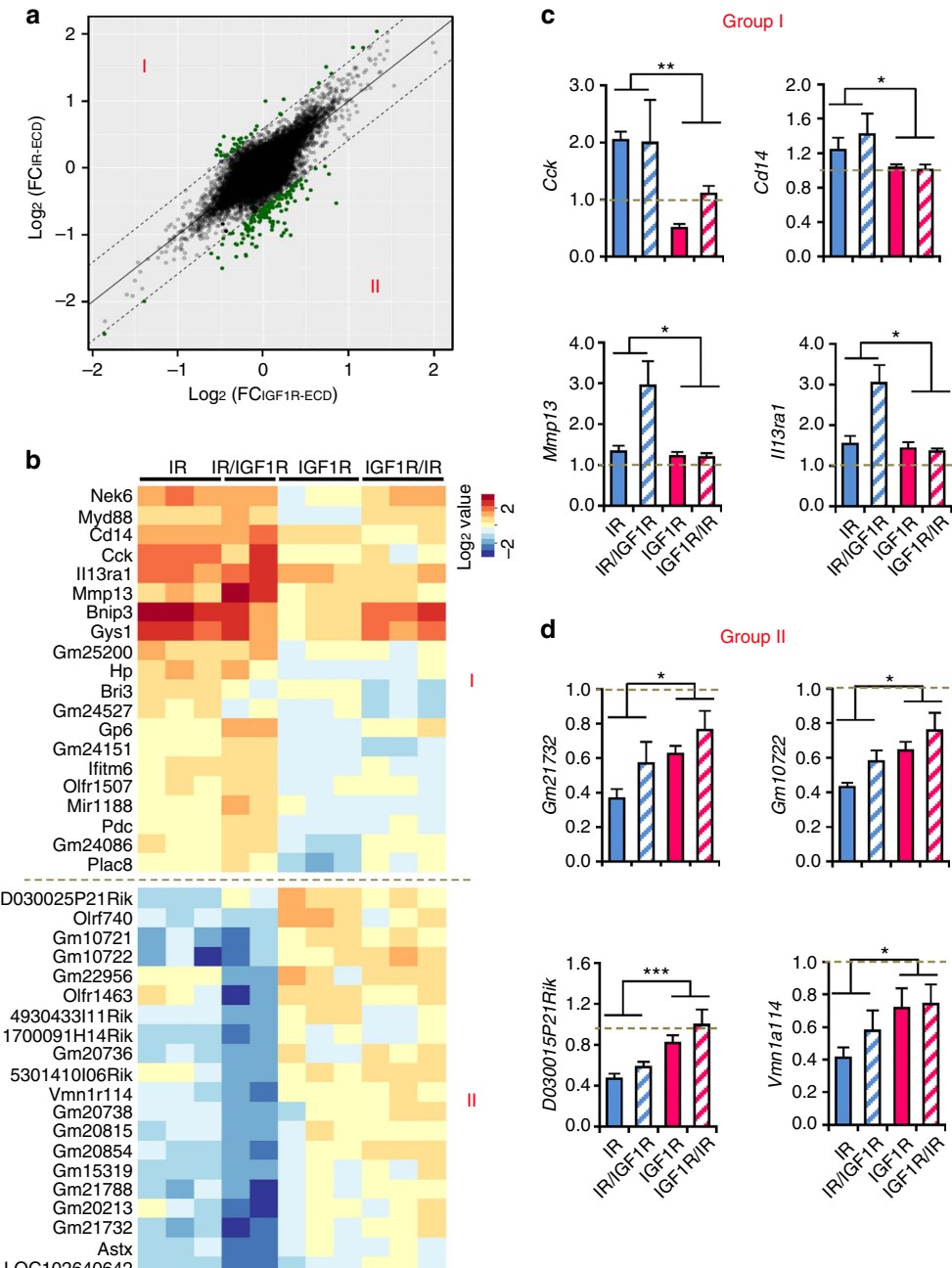

**Figure 4 | Differential roles of IR-ECD and IGF1R-ECD in regulating gene expression.** (**a**) Scattered plot showing distribution of log fold change of each probeset's expression after stimulation in cells expressing receptors with IR-ECD (y axis) versus that in cells expressing receptors with IGF1R-ECD (x axis). Each dot represents the mean of all the clones expressing the receptors with the same ECD. The highlighted dots represent the probe sets with fold-change difference between IR-ECD and IGF1R-ECD over 50%. (**b**) Heat map of top 20 significant genes in both groups in **a**. (**c**) Fold changes of expression in response to stimulation for representative genes highly induced in cells expressing receptors with IR-ECD were confirmed by qPCR versus TBP. (**d**) Fold changes of expression in response to stimulation for representative genes highly suppressed in cells expressing receptors with IR-ECD were confirmed by qPCR versus TBP. Data are mean ± s.e.m. (*P < 0.05; **P < 0.01; ***P < 0.001. One-way ANOVA followed by Newman–Keuls post-hoc analysis, n = 6).

response to insulin stimulation, whereas IR[Y972F] mutation (equivalent to Tyr[960] in IR-A isoform), which has been shown to be required for IRS-1 phosphorylation[22], failed to recruit and phosphorylate either IRS-1 or Shc (Fig. 6b). Replacing Pro[963] and Ser[968] in IR with the corresponding IGF1R residues (Val and Val) had little effect on substrates binding and phosphorylation. By contrast, substitutions of differential residues C terminal to the NPEY motif in IR to corresponding residues in IGF1R (that is, L973F and S976A) resulted in increased Shc binding but decreased IRS-1 binding, especially with the L973F mutation.

To further confirm the substrate preference dictated by the Y + 1 residue in NPEY motif, we performed a Shc competition assay against IRS-1 for binding to IR. While in the absence of Shc, IRS-1 associated with both wild-type IR and mutated IR[L973F] at a similar level, increasing expression of Shc dramatically impaired the association of IRS-1 with IR[L973F] but not with wild-type IR (Fig. 6c, compare lanes 4, 6, 8 to lanes 10, 12, 14). Consistent with this, IR[L973F] associated with Shc to a much higher level than wild-type IR in co-immunoprecipitation assays (Fig. 6d, compare lane 4 to lane 8). These data strongly

indicate that the phenylalanine residue at Y + 1 position in the NPEY motif of IR and IGF1R favors Shc binding over IRS-1.

To explore the structural basis of the substrate preference and the role of the Leu to Phe change at the NPEY + 1 position, we compared the previously reported structures of the Shc and IRS-1

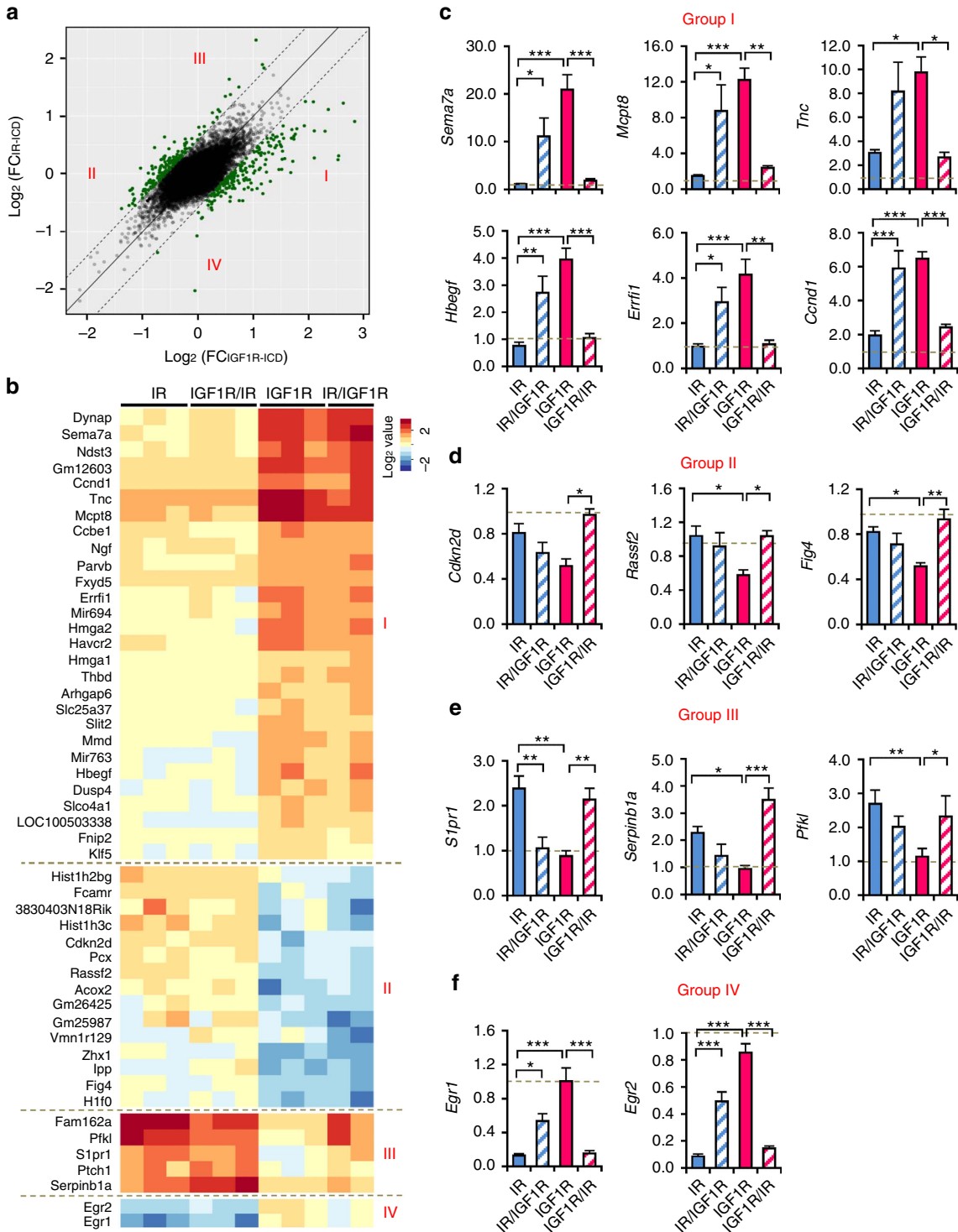

**Figure 5 | Differential roles of IR-ICD and IGF1R-ICD in regulating gene expression.** (**a**) Scattered plot showing distribution of log fold change of each probeset's expression after stimulation in cells expressing receptors with IR-ICD (*y* axis) versus that in IGF1R-ICD (*x* axis). Each dot represents the mean of all the clones expressing the receptors with the same ICD. The highlighted dots represent the probe sets with fold-change difference between IR-ICD and IGF1R-ICD over 50%. (**b**) Heat map of top significant genes in **a** classifying into four groups. (**c,d**) Fold changes of representative genes highly induced (**c**) or suppressed (**d**) by cells expressing receptors with IGF1R-ICD were confirmed by qPCR using TBP as a standard. (**e,f**) Fold changes of expression in response to stimulation for representative genes preferentially induced (**e**) or suppressed (**f**) by cells expressing receptors with IR-ICD were confirmed by qPCR using TBP as a standard. Data are mean ± s.e.m. (*$P<0.05$; **$P<0.01$; ***$P<0.001$. One-way ANOVA followed by Newman–Keuls *post-hoc* analysis, $n = 6$).

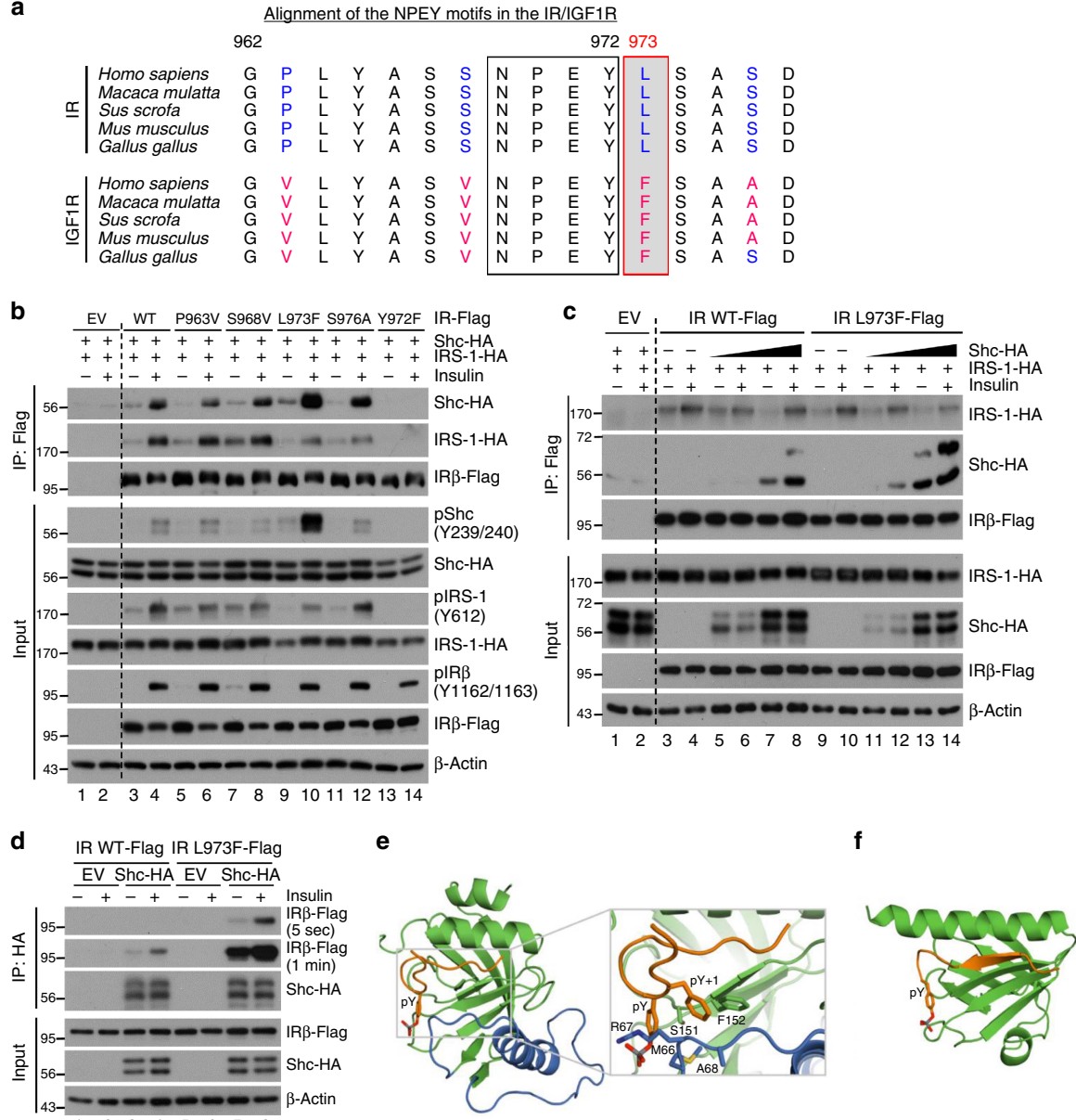

**Figure 6 | Amino acid following NPEY motif dictates the preference of Shc binding. (a)** Alignment of the juxtamembrane regions in the insulin and IGF-1 receptors throughout evolution. **(b)** Co-immunoprecipitation of IR, Shc and IRS-1. Flag-tagged IR or mutated IR were immunoprecipitated with anti-Flag antibody following insulin stimulation. Bound Shc and IRS-1 was detected by biotin-HA immunoblotting. Phosphorylation of proteins from total cell lysates was detected by phospho-specific antibodies. **(c)** Co-immunoprecipitation of IR and IRS-1 with increasing expression levels of Shc. Flag-tagged IR or mutated IR were immunoprecipitated with anti-Flag antibody following insulin stimulation. Bound Shc and IRS-1 was detected by biotin-HA immunoblotting. **(d)** Co-immunoprecipitation of Shc and IR. HA-tagged Shc was immunoprecipitated with anti-HA antibody following insulin stimulation. Bound IR and IR[L973F] were detected by anti-Flag immunoblotting. **(e)** Solution NMR structure of the Shc-PTB domain (green and blue) bound to a phosphorylated segment containing the NPxY motif from the TrkA receptor (brown). Zoom-in: detailed view of the interaction between Shc-PTB and Phe at TrkA Y + 1 position. Note, Phe makes extensive van der Waals contacts in the hydrophobic pocket formed by Phe152, Ser151, Met66 and the aliphatic portion of Arg67. Figure drawn from Protein Data Bank deposition 1SHC. **(f)** Crystal structure of the IRS-1 PTB domain (green) bound to a phosphorylated segment of the juxtamembrane region of the IR (brown). Figure drawn from Protein Data Bank deposition 5U1M.

phosphotyrosine-binding domains (PTB domains) bound to the phosphorylated NPEY motif peptides (Fig. 6e,f and Supplementary Fig. 7). The solution NMR structure of the Shc-PTB domain was determined in complex with a TrkA receptor peptide, which has a Phe at the Y + 1 position like IGF1R[24]. The crystal structure of the IRS-1 PTB domain was determined in complex with a peptide representing the juxtamembrane region of the IR, which has a Leu in the Y + 1 position[25]. Both the Shc and IRS-1 PTB domains bind the

receptor peptide in a generally similar manner. The four to five residues immediately N terminal to the NPEY motif form a β-strand that is sandwiched between the C-terminal α-helix and strand β5 of the PTB domain, while the NPEY motif forms a turn that positions the phosphorylated tyrosine in a positively charged pocket (Fig. 6e). However, unlike the IRS-1 PTB domain, the Shc-PTB domain contains a large insertion that creates a hydrophobic pocket that coordinates the phenylalanine residue in the Y + 1 position. This Y + 1 pocket is formed by Phe152,

Ser[151], Met[66] and the aliphatic portion of Arg[67]. Met[66] and Arg[67] lie within the Shc-specific insertion (Fig. 6e, coloured blue). This arginine is central to correct substrate recognition; in addition to forming part of the $pY + 1$ pocket, the guanidinium group of Arg[67] makes electrostatic contacts with the bound phosphotyrosine. All of these characteristics make the Shc-PTB domain optimal for binding NPEY motifs with a phenylalanine at $Y + 1$ position (Fig. 6e). In contrast, the crystal structure of IRS-1 PTB domain complexed with a phosphorylated NPEY segment of the IR[25] lacks the $Y + 1$ recognition pocket for Phe, while preserving a pY binding pocket and similar cleft for the N-terminal β-strand of the phosphorylated NPEY peptide (Fig. 6f). These observations provide a structural rationale for preferential binding of Shc to IGF1R. The additional binding interactions with the phenylalanine in the $Y + 1$ position should allow it to outcompete IRS-1, which lacks the $Y + 1$ pocket.

**Leu[973] differentiates IR signalling from IGF1R signalling.** To determine if the $Y + 1$ residue in the NPEY motif, which dictates substrate preference, would shift the IR-like signalling and gene expression regulation toward that of the IGF1R, we expressed wild-type IR, IR[Y972F] or IR[L973F] in DKO preadipocytes lacking endogenous IR and IGF1R (Supplementary Fig. 8). Upon insulin stimulation, cells expressing normal IR showed robust activation of all the common downstream signalling pathways, including phosphorylation of IRS-1, Shc, Gab-1, Akt, ERK1/2 and p70S6K1-S6 pathway (Fig. 7a,e and Supplementary Fig. 9). As expected by the lack of substrate binding, IR[Y972F] showed dramatically diminished phosphorylation of all downstream signalling molecules. By contrast, cells expressing IR[L973F] displayed normal receptor phosphorylation (Fig. 7b) and a trend toward decreased IRS-1 and Akt phosphorylation (Fig. 7c,d), but a dramatic increase in Shc and Gab-1 phosphorylation and activation of the ERK1/2 and p70S6K1-S6 pathways compared with cells expressing normal IR (Fig. 7f–j).

Consistent with this, cells expressing IR[L973F] more highly induced multiple IGF1R-ICD upregulated genes, such as *Mcpt8*, *Hbegf*, *Sema7a*, *Tnc* and *Ccnd1* (Fig. 7k, left), and more highly suppressed several IGF1R-ICD suppressed genes, such as *Fig4*, *Rassf2* and *Cdkn2d* (Fig. 7l, left). In addition, some genes, such as *Pfkl*, *Egr1* and *Egr2*, shifted from IR-ICD up or downregulated to non-regulated in cells expressing IR[L973F] (Fig. 7k, right and Fig. 7l, right). However, not all of the gene responses in the cells expressing IR[L973F] showed the same pattern as those in chimeric receptors. For instance, *Dynap*, which was highly selectively stimulated in cells expressing receptors with IGF1R-ICD, was not induced in IR[L973F] cells (Supplementary Fig. 10a), and sequence similarity 162, member A (*Fam162a*) was equally upregulated in wild-type IR and IR[L973F] (Supplementary Fig. 10b). Taken together, these data show that a single amino acid difference of Leu to Phe at position 973 in the juxtamembrane domain of the IR versus IGF1R is sufficient to regulate downstream substrate phosphorylation preference and many of the differential gene expression responses between receptors with an IR-ICD versus receptors with an IGF1R-ICD.

## Discussion

Insulin and IGF-1 receptors are highly homologous and engage similar signalling pathways, yet at a physiological level these receptors exert very different effects. IR largely controls metabolism, whereas IGF-1 receptor controls growth. The molecular mechanisms underlying the distinct functions of IR and IGF1R, however, remain largely unknown. In the present study, we have used cells that express only one receptor isotype, thereby eliminating the possible ligand/receptor cross-activation

and the existence of the IR/IGF1R hybrid receptors. This system provides a clean platform to investigate the differential roles of IR and IGF1R in cells. With this system, we show that the ICDs, and to a lesser extent the ECDs, of these two receptors create differences in the activation of different intracellular signalling pathways leading to distinct regulations of gene expression. Point mutagenesis demonstrates that a major determinant of this differential signalling is a single residue difference in the intracellular juxtamembrane region just C terminal to the NPEY motif of IR and IGF1R (Leu in IR and Phe in IGF1R).

IR and IGF1R are known to phosphorylate multiple IR substrates, including IRS-1, IRS-2, Shc, Gab-1 and Grb10 (ref. 26). Our data clearly show that the ICD of the IR is more effective in phosphorylating IRS-1, whereas receptors with the IGF1R-ICD interact with and phosphorylate more potently in Shc. These findings are supported by phosphotyrosine proteomic studies showing stronger phosphorylation of IRS-1 than Shc following insulin stimulation[27].

These findings are also supported by structural data. The ICDs of IR and IGF1R contain the juxtamembrane region immediately after the transmembrane helix, followed by the highly conserved tyrosine kinase domain and a flexible C-terminal tail. The kinase domains of IR and IGF1R share almost identical overall structures[28,29] and are responsible for receptor autophosphorylation and tyrosine phosphorylation of the substrates[30]. The less conserved juxtamembrane and C-terminal regions of IR and IGF1R provide docking sites for many substrates and adaptor proteins upon tyrosine phosphorylation. The juxtamembrane region, especially the phosphorylated tyrosine at position 972 in the NPEY motif of IR has been shown to be important for IRS and Shc recruitment[22,23]. Mutation of Tyr[972] in the NPEY motif of the IR virtually abolishes IRS-1 phosphorylation and the activation of downstream signalling pathways[22]. A similar NPxY motif is found in many other transmembrane receptors including the EGF receptors, TrkA receptor and receptors with no known kinase activity like the LDL receptor. The NPxY motif (NPEY in IR and IGF1R) is not only critical for substrate recruitment, but also required for rapid internalization of the receptors, at least in IR, IGF1R and LDL receptor[31–33].

Although the NPEY motifs in both IR and IGF1R are able to bind IRS-1 and Shc, our study shows the amino acid differences adjacent to the NPEY motif play a critical role in substrate preference. Thus, replacement of leucine[973] C terminal to the NPEY motif in the IR by phenylalanine, the residue found in the homologous region of the IGF1R, is sufficient to largely switch the cell signalling and gene expression patterns toward those of the IGF1R. Cells expressing IR[L973F] display not only a trend toward reduction in IRS-1 phosphorylation, but also dramatically higher phosphorylation of Shc, mimicking the pattern observed in cells expressing receptors with IGF1R or the chimeric IR/IGF1R. This is due to the stronger binding of Shc to IR[L973F] than wild-type IR upon insulin stimulation. This is further supported by structural data that reveal a larger hydrophobic pocket in the Shc-PTB domain, which is absent in IRS-1 PTB domain, making Shc more suitable for binding NPEY motifs with Phe at $Y + 1$ position[24,25]. This is also supported by point mutation studies, which have shown that alanine substitutions of residues N terminal to the NPEY motif in IR, such as Leu[964] and Tyr[965] disrupt the IRS-1 interaction without affecting Shc binding, whereas substitute of Leu[973] by alanine severely reduces Shc binding with little effect on IRS-1 binding[34]. Thus, while the Tyr residue in NPEY motif is essential for substrate recruitment, it is the residues surrounding the NPEY motif that dictate the substrate preference, and a key residue in these interactions is the Leu versus Phe at $Y + 1$ position after NPEY motif.

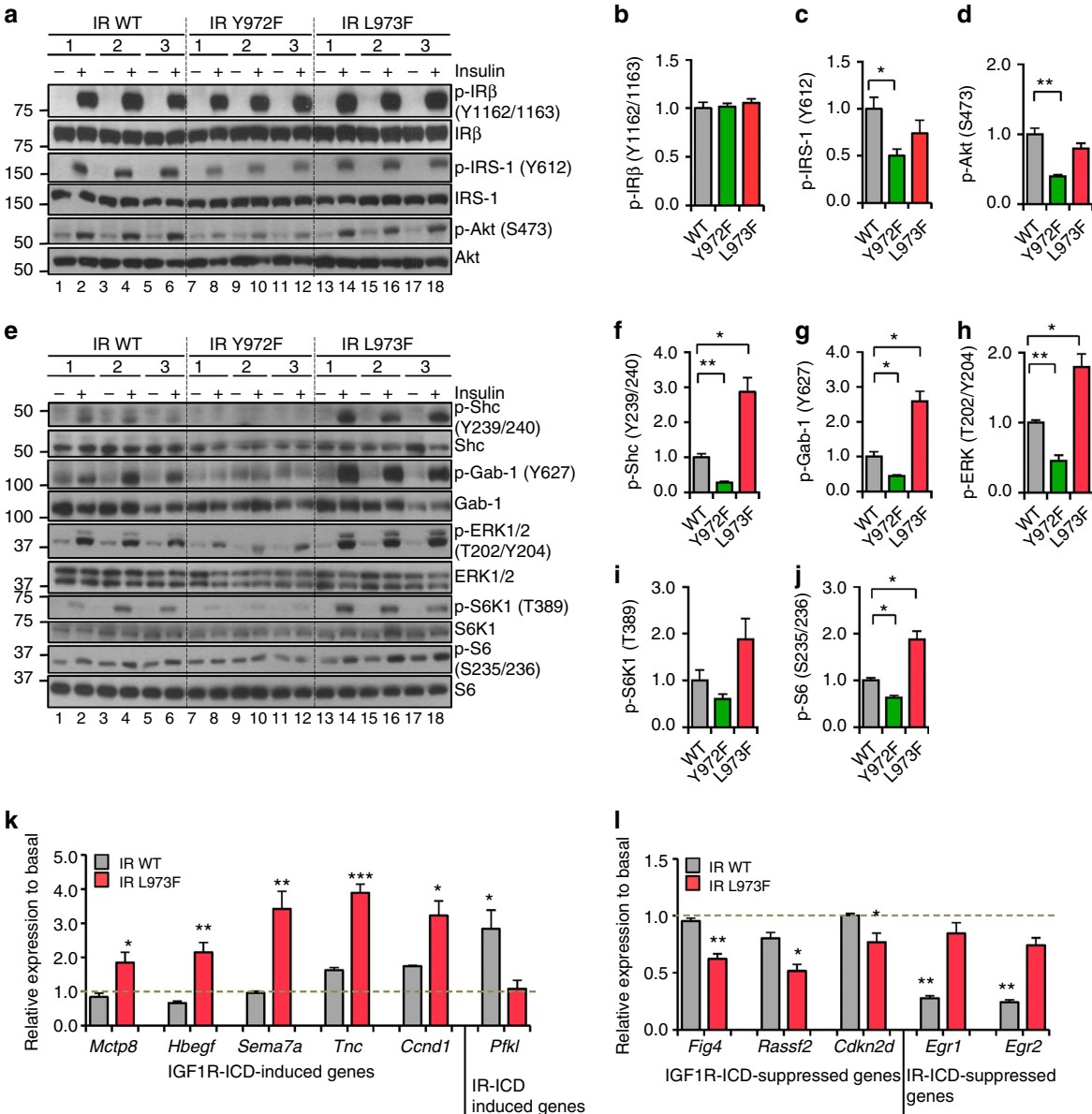

**Figure 7 | Leu$^{973}$ is a key residue differentiating IR signalling from IGF1R signalling.** (**a**) Immunoblotting of the phosphorylation of IR, IRS-1 and Akt in lysates from preadipocytes expressing normal IR, IR$^{Y972F}$ and IR$^{L973F}$ stimulated with 100 nM insulin for 5 min. (**b–d**) Densitometric analysis of phosphorylated IR, IRS-1 and Akt following 5 min stimulation. Data are mean ± s.e.m. (*$P < 0.05$; **$P < 0.01$. One-way ANOVA followed by $t$-test with Bonferroni correction. $n = 3$). (**e**) Immunoblotting of the phosphorylation of Shc, Gab-1, ERK, p70S6K1 and S6 in lysates from preadipocytes expressing normal IR, IR$^{Y972F}$ and IR$^{L973F}$ stimulated with 100 nM insulin for 5 min. (**f–j**) Densitometric analysis of phosphorylated Shc, Gab-1, ERK, p70S6K1 and S6 following 5 min stimulation. Data are mean ± s.e.m. (*$P < 0.05$; **$P < 0.01$. One-way ANOVA followed by $t$-test with Bonferroni correction. $n = 3$). (**k,l**) Fold changes of expression in response to stimulation for representative genes regulated by normal IR or IR$^{L973F}$ were confirmed by qPCR with TBP as the housekeeping gene. Data are presented as mean ± s.e.m. (*$P < 0.05$; **$P < 0.01$; ***$P < 0.001$. Student $t$-test. $n = 3$).

Interestingly, the phosphorylation patterns of the downstream kinases ERK1/2 and S6K1 parallel Shc phosphorylation, which itself follows differences in the ICDs of the IR and IGF1R, especially at the Y + 1 residue in the NPEY motif. In contrast, Akt phosphorylation is comparable among all four types of cells, whether expressing receptors with IR or IGF1R ICDs. Future studies with more comparisons of the complete phosphoproteome of these cells should help further dissect the signalling networks regulated by the ICDs of IR and IGF1R.

Insulin and IGF-1 are potent hormones in control of gene transcription. However, previous attempts to map the differential expression profiles regulated by insulin and IGF-1 signalling[15,16] have been complicated by the fact that most cells contain both

endogenous IR and IGF1R, and the fact that both the receptors are capable of binding both ligands, especially at the high concentrations used in many *in vitro* experiments. Here, we are able to assess regulation of gene expression in the cells with only one type of receptor and all at similar levels of expression. For many experiment, we also used 10 nM of ligand for stimulation, which is a more physiological level than the 100 nM used in many previous experiments. As expected, a significant portion of genes are regulated by both IR and IGF1R, however, many genes are differentially regulated, demonstrating the distinct, but overlapping roles of IR and IGF1R in regulation of gene expression.

These differences in gene regulation are mainly dependent on the ICDs of the receptors, consistent with the major differences in

intracellular signalling regulation, especially Shc phosphorylation, depending on the ICD of the receptors. This suggests that Shc-dependent regulation of gene expression might be responsible for a significant fraction of IGF1R-ICD-dependent gene regulation. Besides the previously reported genes, like *Hbegf*[35], *Egr1* and *Egr2* (refs 15,17), we have also identified multiple new differentially regulated genes by the IR or IGF1R ICDs. Some of these newly identified genes show striking 10–15-fold differences in regulation between receptors with IR-ICD and IGF1R-ICD (for example, *Sema7a*, *Mcpt8*, *Tnc*). We also identify several microRNAs and long non-coding RNAs that are differentially regulated by IR and IGF1R, some of which have been reported to be important in brown adipocyte differentiation and function[36,37].

One interesting finding from our study is that the ECDs of IR and IGF1R also contribute to differential signalling events (that is, receptor internalization and signalling kinetics) and even the expression of some genes in response to stimulation. Exactly how this occurs will require further study, but at least three factors could contribute. Firstly, structural analysis and alanine mutagenesis studies have revealed that insulin and IGF-1 interact with the different regions of the ECD of the receptors[38–41], creating different 3-dimentional configurations. The high affinity binding site (site 1) of IR for insulin locates in the L1 domain and part of the second fibronectin type 3 domain ($Fn_2$)[38], whereas the same binding site in IGF1R for IGF-1 is extended into the cysteine-rich regions from L1 domain, which is responsible for the interaction of C-peptide of IGF-1 (ref. 42), which is cleaved and absent in insulin. Thus, these differences contribute to the different ligand binding kinetics between IR and IGF1R[43], and might be responsible for some of the differences in signalling and gene expression regulation by IR and IGF1R. Secondly, the three primary ligands of this receptor system—insulin, IGF-1 and IGF-2—may interact differently even with the same receptor ECDs, and this can contribute to differential signalling even when interacting with the same receptor[44]. Finally, the ECD of the receptors interacts with other membrane or extracellular proteins, creating a unique pattern of actions. Indeed, insulin, but not IGF-1 receptor, has been shown to associate with GPI-anchored glypican 4 (GPC4). This IR/GPC4 interaction alters receptor binding and sensitizes IR signalling[45]. In addition, the transmembrane domains of IR and IGF1R may also contribute to some of the signalling differences in our chimeric receptor system, since transmembrane domains of receptor tyrosine kinases, especially IR, have been shown to play an important role in signal propagation in response to ligand binding[46].

In summary, we have demonstrated that IR and IGF1R have similar, but distinct, patterns of regulation of intracellular signalling, inducing effects on gene expression. While ECD differences have important effects on ligand binding and some signalling events, the major differences in insulin and IGF-1 action are due to differences in the juxtamembrane ICDs of their receptors, especially in sequence difference at position 973 in this region of the IR and 951 in the IGF1R. These intrinsic receptor differences contribute to differences in substrate phosphorylation between these hormones and in ability to exert metabolic control versus mitogenic regulation. These findings help explain the fundamental differences in an IR and IGF1R signalling, and will provide an opportunity for differentially targeting these pathways at a pharmacological level.

## Methods

**Materials.** Mouse IR (MC224356) and IGF1R (MC224342) cDNA clones were purchased from Origene. Wild-type IR and IGF1R as well as chimeric receptors IR/IGF1R (IR-ECD (aa 1–919) fused to IGF1R transmembrane and ICD (aa 908–1,339), numbers excluding signal peptide) and IGF1R/IR (IGF1R-ECD

(aa 1–907) fused to IR transmembrane and ICD (aa 920–1345)) were subcloned into the pBabe-hyrgromycin vector. To generate chimeric receptors, the Ile[947] to Leu point mutation was introduced into the IR cDNA to generate a BclI restriction site using the primer pair (5′-ccatcaaatattgccaaactgatcattggacccctcatc-3′; IR BclI 3: 5′-gatgaggggtccaatgatcagtttggcaatatttgatgg-3′). The human IR-B isoform retroviral plasmid was generated previously in the lab[17]. IR[P963V], IR[S968V], IR[Y972F], IR[L973F], IR[S976A] (aa numbers excluding signal peptide) were generated from the human IR-B isoform cDNA using a site-directed mutagenesis kit from Agilent. Primer pairs for site-directed mutagenesis were as follows:

IR[P963V]: 5′-gaagcgtaaagcactcccagcggcccatctg-3′ and 5′-cagatgggccgctgggagtgcttt acgttc-3′;

IR[S968V]: 5′-actgagatactcagggtttacagaagcgtaaagcggtccc-3′ and 5′-gggaccgctttacgc ttctgtaaacccctgagtatctcagt-3′;

IR[Y972F]: 5′-ttacgcttcttcaaaccctgagtttctcagtgccag-3′ and 5′-ctggcactgagaaactcaggg tttgaagaagcgtaa-3′;

IR[L973F]: 5′-cactggcactgaaatactcagggtttgaagaagcgt-3′ and 5′-acgcttcttcaaaccct gagtatttcagtgccagtg-3′;

IR[S976A]: 5′- gagcatggaaacacatcagcggcactgagatactcagg-3′ and 5′- cctgagtatctcagt gccgctgatgtgtttccatgctc-3′.

For co-immunoprecipitation assays, human IR, IR[Y972F] and IR[L973F] cDNA were cloned into 3 × Flag-CMV-14 mammalian expression vector (Sigma). The cDNA's for human Shc (NM_003029) and mouse IRS-1 (NM_010570) were cloned into the pKH3 vector to generate c terminal, HA-tagged expression constructs.

Antibodies against phospho-IR/IGF1R (#3024, 1:500), IGF1Rβ (#3027, 1:500), phospho-Shc (Y239/240) (#2434, 1:500), Shc (#2432, 1:500), phospho-Gab-1 (Y627) (#3233, 1:500), Gab-1 (#3232, 1:500), phospho-ERK1/2 (T202/Y204) (#9101, 1:1,000), ERK1/2 (#9102, 1:1,000), phospho-Akt (S473) (#9271, 1:1,000), Akt (#4685, 1:1,000), phospho-S6K1(T389) (#9205, 1:1,000), phospho-S6 (S235/236) (#2211, 1:2,000), S6 (#2317, 1:500), GAPDH (#5174, 1:1,000) were purchased from Cell Signaling Technologies. Anti-β-Actin (sc-1616-HRP, 1:10,000), IRβ (sc-711, 1:500) and p70 S6K1 (sc-230, 1:500) antibodies were from Santa Cruz. Phospho-IRS-1 (Y612) (09-432, 1:1,000) antibody was purchased from Millipore. Anti-IRS-1 (611394, 1:500) antibody was from BD Biosciences. Human insulin was purchased from Sigma and human IGF-1 from Preprotech.

**Brown preadipocytes isolation and culture.** All animal studies were approved by the Institutional Animal Care and Use Committee (IACUC) at the Joslin Diabetes Center and were in accordance with the National Institutes of Health guidelines. Preadipocytes were isolated from newborn IR-lox/IGF1R-lox mice by collagenase digestion of brown fat and immortalized by infection with retrovirus encoding SV40 T-antigen followed by the selection with $2\,\mu g\,ml^{-1}$ of puromycin. The immortalized preadipocytes were infected with adenovirus containing GFP alone (to generate control cell line) or GFP-tagged Cre recombinase. GFP-positive cells were sorted by FACS and expanded in DMEM supplemented with 10% heat-inactivated fetal bovine serum (FBS, Sigma), $100\,U\,ml^{-1}$ penicillin and $100\,\mu g\,ml^{-1}$ streptomycin (Gibco) at 37 °C in a 5% $CO_2$ incubator[17].

IR/IGF1R double knockout preadipocytes were then stably transduced using the pBabe retrovirus system to generate mouse IR, IGF1R, IR/IGF1R IGF1R/IR chimeric receptor, human IR, IR[Y972F] or IR[L973F] cell lines. Briefly, human embryonic kidney 293T cells (ATCC) were transiently transfected with 10 μg of the pBabe-hygro retroviral expression vectors encoding wild-type or mutant IR or IGF1R sequences and viral packaging vectors SV-E-MLV-env and SV-E-MLV using TransITExpress transfection reagent (Mirus Bio). 48 h after transfection, virus-containing medium was collected and passed through a 0.45 μm pore size syringe filter. Polybrene (hexadimethrine bromide; $12\,\mu g\,ml^{-1}$) was added and the medium was applied to proliferating (40% confluency) DKO cells. 24 h after infection, cells were treated with trypsin and re-plated in a medium supplemented with hygromycin (Invitrogen). Cells were maintained in DMEM supplemented with 10% FBS, $100\,U\,ml^{-1}$ penicillin and $100\,\mu g\,ml^{-1}$ streptomycin (Gibco), and cultured at 37 °C in a humidified atmosphere of 5% $CO_2$.

**Fluorescence-activated cell sorting.** To achieve equivalent expression of the receptors, a 2-step fluorescence-activated cell sorting (FACS) approach was conducted using auto-antibodies against IR from a patient with severe insulin resistance, which also cross-reacted with IGF1R. Overall, $1 \times 10^6$ cells were gently detached using 1 × Accutase (ThermoFisher Scientific) and washed with 1 × phosphate buffer saline (PBS). Cells were stained with patient serum (containing auto-antibodies) diluted 1:500 in PBS with 2% horse serum for 1 h at RT, followed by washing with 1 × PBS and incubation with anti-human Alexa Fluor 488 for 30 min at 4 °C. After three washes with 1 × PBS, cells were analysed and sorted to isolate cells with equivalent expression using a FACSAria cell sorter and re-plated on 10 cm plates for further experiments. To test for cell viability, $0.1\,mg\,ml^{-1}$ of propidium iodide (Sigma-Aldrich; St Louis, MO, USA) was added to the cells 1–2 min prior to analysis. After 2 weeks in culture, the same FACS approach was performed to perform a second round of enrichment of cell lines exhibiting equivalent levels of receptor expression. Subsequently, single cell clones were isolated, expanded and maintained for analysis. The receptor expression in each clone was confirmed by qPCR and immunoblot analysis.

**Proliferation assay.** Cells expressing normal IR, IGF1R, or chimeric receptors IR/IGF1R and IGF1R/IR were seeded into four 35 mm dishes (50,000 cells per dish) at day 0 and cultured in Dulbecco's modified Eagle's medium (DMEM) + 10% fetal bovine serum (FBS, Sigma). Every 24 h, one dish of each cell line was trypsinized, resuspended and counted for the total cell number. Doubling times per 24 h for each cell line were calculated.

**Glycolytic stress test.** The glycolysis rate of brown preadipocytes were measured by Seahorse XF24 Bioanalyzer (Seahorse Bioscience). Briefly, cells were seeded into XF24 cell culture microplates at the density of 60,000 cells per well and serum starved overnight followed by 100 nM insulin or IGF-1 stimulation for 6 h. Before analysis, cells were washed with warm $1 \times$ PBS once, incubated in 550 μl basal XF assay media (DMEM (Sigma) without pyruvate and glucose supplemented with $1 \times$ Glutamax (Sigma)) and incubated at 37 °C without $CO_2$ for 1 h. ECAR values were measured using a Seahorse XF24 Bioanalyzer according to manufacturer's glycolytic stress test protocol. Briefly, ECAR values were measured in the absence of glucose, after injection of glucose (final concentration: 10 mM), after injection of oligomycin (final concentration: 2 μM) and after injection of 2-deoxy-glucose (final concentration: 50 mM). Cells were lysed in 80 μl 0.1% SDS solution and protein concentrations were measured and used for normalization of ECAR values. The basal glycolysis rate was calculated by comparing ECARs before and after glucose injection. The maximal glycolytic capacity was calculated by subtracting ECAR value after oligomycin injection from the ECAR in the absence of glucose.

**Insulin and IGF-1 signalling.** Cells were serum starved for 5 h with DMEM containing 0.1% BSA. Cells expressing normal IR, chimeric receptor IR/IGF1R, or IR with single amino acid mutations ($IR^{Y972F}$ and $IR^{L973F}$) were stimulated with 10 or 100 nM insulin for indicated times, while cells expressing normal IGF1R or chimeric receptor IGF1R/IR were stimulated with 10 or 100 nM IGF-1 for indicated times. After stimulation, cells were washed immediately with ice-cold PBS once before lysis and scraped down in RIPA lysis buffer complemented with 50 mM KF, 50 mM β-glycerolphosphate, 2 mM EGTA (pH 8), 1 mM $Na_3VO_4$ and $1 \times$ protease inhibitor cocktail (Calbiochem). Protein concentrations were determined using the Pierce 660 nm Protein Assay Reagent (Bio-Rad). Lysates (10–20 μg) were resolved on SDS-PAGE gels, transferred to PVDF membrane for immunoblotting.

**Receptor internalization assay.** Cells expressing wild-type IR and IGF1R or chimeric receptors (IR/IGF1R or IGF1R/IR) were serum starved in DMEM supplemented with 0.1% BSA for 3 h, followed by the stimulation with 100 nM insulin (for IR and IR/IGF1R) or 100 nM IGF-1 (for IGF1R and IGF1R/IR) for 0, 30 and 120 min. The cells were rinsed once with ice-cold PBS, followed by 1 ml 0.3 mg ml$^{-1}$ sulfo-NHS-Biotin (ThermoFisher Scientific) labelling at 4 °C for 30 min, and the labelling reaction was quenched by the addition of ice-cold 100 mM glycine (pH 3) for 10 min. Cells were washed twice with ice-cold PBS, and lysed in RIPA buffer supplemented with 10 mM glycerophosphate, 10 mM NaF, 0.1 mM sodium orthovanadate and 1% protease inhibitor cocktail (Sigma). Biotinylated surface proteins were enriched by incubating 100 μg total protein lysates with 10 μl streptavidin-agarose beads suspension (50% slurry, ThermoFisher Scientific) in 800 μl total volume on a rotator at 4 °C for 1 h. Subsequently, beads were washed with RIPA lysis buffer three times and bound proteins were liberated from the beads by boiling in $1 \times$ SDS-PAGE loading buffer. The biotinylated surface protein fractions as well as total protein lysates were resolved on SDS-PAGE gels, transferred to PDVF membrane and immunoblotted using IRβ and IGF1Rβ antibodies.

**Transfection.** HEK-293T (ATCC) cells were transfected using Superfect transfect reagent (Qiagen) according to the manufacturer's protocol for the co-immunoprecipitation assays.

**Co-immunoprecipitation.** To examine protein interactions, cell lysates were prepared from HEK-293T cells transiently co-transfected with epitope-tagged protein expression vectors (Flag-EV, IR WT-Flag, $IR^{P963V}$-Flag, $IR^{S968V}$-Flag, $IR^{L973F}$-Flag, $IR^{S976A}$-Flag, $IR^{Y972F}$-Flag, HA-EV, Shc-HA as indicated). Forty-eight hours after transfection, cells were serum starved for 5 h in DMEM + 0.1% BSA, followed by 5 min stimulation of 10 nM Insulin at 4 °C to minimize ligand-stimulated internalization and degradation. Total cell lysates were prepared using lysis buffer (20 mM Hepes (pH 7.4), 150 mM NaCl, 50 mM KF, 50 mM β-glycerolphosphate, 2 mM EGTA (pH 8), 1 mM $Na_3VO_4$, 1% Triton X-100, 10% glycerol and $1 \times$ protease inhibitor cocktail (Calbiochem)). Overall, 400 μg protein lysates were incubated with 10 μl anti-HA magnetic beads (Pierce) in a total volume of 800 μl for 1 h at 4 °C with end-to-end rotation. The immunocomplexes were washed sequentially: 1 time with lysis buffer, two times with lysis buffer + 500 mM NaCl, and two times with lysis buffer. Bound proteins were eluted by incubation for 5 min at 100 °C in $1 \times$ SDS loading buffer. The bound proteins along with 10 μg total cell lysates from each sample were resolved using SDS-PAGE, transferred to PVDF membranes and subjected to immunoblotting using the indicated antibodies including biotinylated-HA antibody for HA-tagged Shc detection.

**Immunoblotting.** Membranes were blocked in Starting Block T20 (ThermoFisher) at room temperature for 1 h, incubated with the indicated primary antibody in Starting Block T20 solution overnight at 4° C. Membranes were washed three times with 1X PBST, incubated with HRP-conjugated secondary antibody (GE Healthcare, anti-mouse IgG, NA931; anti-rabbit IgG, NA934; 1:20,000) in Starting Block T20 for 1 h and signals detected using Immobilon Western Chemiluminescent HRP Substrate (Millipore). Original scans of full-size membrane strips probed with the respective antibodies are presented in Supplementary Figs 11–14.

**Insulin and IGF-1 stimulation for RNA isolation.** Cells were serum starved overnight with DMEM + 0.1% BSA. Cells were then mock treated or treated with insulin or IGF-1 (100 nM) for 6 h and then washed once with cold $1 \times$ PBS and resuspended in RLT lysis buffer (Qiagen). Total RNA was extracted using an RNeasy mini kit (Qiagen) following manufacturer's manual.

**Microarray and bioinformatics analysis.** Microarrays were processed using the WT PLUS kit and Mouse Gene 2.0 ST arrays from Affymetrix. Total RNA input was 250 μg. For the final step, 5.5 μg cDNA were fragmented and labelled, and the chips were hybridized for 16 h in the GeneChip Hybridization Oven 645 using 3.5 μg of this fragmented and labelled product. Chips were then scanned using the GeneChip Scanner 7,000, and normalized using RMA and Affymetrix's AGCC software. Then bioinformatic analysis was done in R/Bioconductor[47] examining paired differences between stimulated versus unstimulated samples. One pair was an extreme outlier in the principal component analysis plot and received a low quality weight[48], and thus was removed. However, all the cell lines were included for the further qPCR confirmation. Fold change of the expression of each probeset was calculated by comparing the gene expression after 6 h insulin or IGF-1 stimulation with the 6 h mock treated cells. Statistical significance of probe sets was assessed with empirical Bayesian linear modelling using the limma package[49], and significance of gene sets was assessed with the sigPathway package[50]. Heatmaps were created with the gplots package, and volcano plots and scatterplots were created with the ggplot2 package[51].

**Analysis of gene expression by quantitative PCR.** Overall, 1 μg of RNA was reverse transcribed using a High Capacity cDNA Reverse Transcription kit (Applied Biosystems) according to the manufacturer's instructions. Real time PCR (Supplementary Table 7) was performed using the SYBR Green PCR master mix (Bio-Rad). Fluorescence was monitored and analysed in an ABI Prism 7900 HT sequence detection system (Applied Biosystems). TBP expression was used to normalize gene expression. Amplification of specific transcripts was confirmed by analysing melting curve profiles at the end of each PCR.

**Data availability.** The NMR structural data of Shc-PTB domain was downloaded from RCSB protein data bank with the PDB ID 1SHC. The crystal structural data of IRS-1 PTB domain was downloaded from RCSB protein data bank with the PDB ID 5U1M. The Affymetrix microarray data generated in this study are available at NCBI GEO database with the accession number GSE81921. All the other data and original codes used in this study are available from the corresponding author on reasonable request.

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

## Acknowledgements

This work was supported by NIH Grants R01 DK031036 and R01 DK033201 (to C.R.K.) and Sanofi. M.S. was supported by Manpei Suzuki Diabetes Foundation and a fellowship from the Japan Society for the Promotion of Science. A.K. was supported by a German Research Foundation (DFG) fellowship Kl2399-1/1 and 4/1 grant and by the Federal Ministry of Education and Research (German Center for Diabetes Research, Grant No 01GI092). A.K.R. was supported by NIH Grant T32 DK007260-37. B.T.O. was supported by NIH Grant K08 DK100543. The Joslin Diabetes Center DRC Genomics and Bioinformatics Core (P30 DK36836) also provided important help.

## Author contributions

W.C., M.S. and A.K. designed the study, researched data and wrote the manuscript. G.G.-D.P., J.M.D., B.T.O., A.K.R., H.P., J.N.W., J.B. and M.J.E. researched data and helped design experiments. C.R.K. designed the study, supervised all work and helped write the manuscript.

## Additional information

**Competing interests:** The authors declare no competing financial interests.

