## [Peer Review File · Nature Communications]

Reviewer #1 (Remarks to the Author)

Insulin through binding to the insulin receptor (IR) and IGF-I and II binding to the IGF-I receptor (IGF1R) elicit distinct biological effects (primarily but not exclusively metabolic for insulin and primarily but not exclusively mitogenic for the IGFs). The structural basis of this signaling specificity has remained mysterious given the high degree of sequence homology between the two receptors, and the largely overlapping intracellular signaling pathways elicited by the receptor kinase activation. The problem is compounded by the existence of hybrid IR/IGF1R in cells that have both types of receptors.

In this exciting paper, Cai, Sakaguchi et al. use an elegant approach to solve this conundrum. They generated brown preadipocytes in which both endogenous IR and IGF1R have been genetically inactivated using Cre-lox recombination (IR and IGF1R DKO cells). These DKO cells were then reconstituted with either wild-type mouse IR, IGF1R, or one of two chimeric receptors: IR/IGF1R with the IR extracellular domain (ECD) fused to the IGF1R transmembrane and intracellular domain (ICD) and IGF1R/IR with the ECD of IGF1R fused to the transmembrane and ICD of IR. These cell lines will have no hybrid receptors.

The levels of receptor expression was similar in all cell lines. The ligand-dependent activation of the receptors was performed using the ligand matching the ECD of the overexpressed receptor. All receptors showed a robust and similar level of receptor autophosphorylation.

The mitogenic defect observed in the background DKO cells was rescued by the wild-type IGF1R, but less well by the IR. The constructs containing the IGF1R ICD showed a greater mitogenic potential than those with IR ICD. The opposite was true for metabolic activity.

Receptors with IR ECD showed more rapid internalization.

There were also marked differences in downstream targets phosphorylation by the four receptors. The intracellular domain of IR was more potent in phosphorylating IRS-1 while the ICD of IGF1R was better in activating Shc, Gab-1 and p70S6K1 signaling pathways.

The authors also studied the gene expression patterns elicited by the four receptors and showed marked differences depending on both the ECDs and ICDs, with the IR domains being more associated with metabolic signaling.

Finally, the authors examined the role of the juxtamembrane domain NPEY motif and showed that it confers the observed specificity of the ICD, and by site-directed mutagenesis showed that substituting Leu 973 to Phe (like in IGF1R) in the IR ICD shifts the signaling pattern to that of the IGF-1R, showing that this residue plays a critical role in substrate preference. The authors propose a structural interpretation based on the NMR structure of the Shc PTB domain bound to a phosphorylated NPEY domain of the TrkA receptor.

The overall major conclusion is that the differences in insulin and IGF-I action are due to differences in the juxtamembrane ICDs of their receptors, especially in sequence difference at position 973 in the IR and 951 in the IGF1R. These intrinsic receptor differences contribute to differences in substrate phosphorylation between these ligands and in their ability to exert metabolic control versus mitogenic regulation.

The article is convincing and well written, and nicely illustrated. The experimental approach is logical and straightforward, and complex experiments are presented in an accessible way. The work brings a novel and important element for discussion of the molecular and structural basis of intracellular signaling specificity.

One has to remain aware though that this elegant reductionist approach markedly simplifies the system by having cell lines with only one receptor or receptor chimera stimulated by one ligand, while in real life most cells will have both IR and IGF1R plus hybrids, being stimulated by three ligands, insulin, IGF-I and IGF-II. The combinatorial nature of signaling under such complex conditions remains to be explored.

I have only one substantial critique and some very minor ones.

A weak part of the paper is the discussion of the role of the extracellular domain of IR and IGF1R in differential signaling. The authors state the following:

"The IR extracellular domain harbors two interaction domains, one with high affinity and the other one with low affinity. Binding for insulin, providing unique binding kinetics for negative cooperativity. By contrast, the IGF1R has only one IGF-I binding surface consisting of the cysteine-rich region and the flanking regions from L1 and L2 domains".

The statement regarding the IGF1R having only one binding surface is incorrect and ignores most of the recent advances in the structural biology of IGF1R binding. Firstly, the IGF1R also exhibits negative cooperativity in binding, suggesting a similar binding mode to that of insulin (Christoffersen CT et al., *Endocrinology* 135:4772-4775, 1994). One difference is that the IGF1R dose-response curve is not bell-shaped. Secondly, a second binding site homologous to insulin's binding site 2 has been identified on IGF-I by alanine scanning mutagenesis, made of Glu9, Asp12, Phe16, Leu54 and Glu 58. Multiple substitutions in this site 2 result in a 33 to 100-fold reduction in affinity for the IGF1R (Gauguin L et al., *J Biol Chem* 283:20821-20829, 2008). Likewise, site 2 on IGF-II has been mapped (Albino CL et al., *J Biol Chem* 284:7656-7664, 2009). Thirdly, mathematical modeling of insulin and IGF1R binding kinetics has quantitatively accounted for all the kinetic and equilibrium binding properties of both IR and IGF1R based on the same bivalent crosslinking model (Kiselyov VV et al., *Mol Syst Biol* 5:243, 2009). The authors are correct in that for the IGF1R site 1 extends into the cysteine-rich region which has been previously shown to accommodate the C-domain absent in insulin.

One point not discussed by the authors which may deserve mention is that the ligands themselves contain structural information that has some role in determining differential signaling through the same receptor. It has been shown in cells only expressing IGF1R that insulin, IGF-I and IGF-II elicit overlapping but non-identical patterns of gene expression, with some genes regulated by only one of the ligands (*Front Endocrinol (Lausanne)* 4:98, 2013). This is biologically highly relevant since unlike Vertebrates, Invertebrates have only one insulin-like receptor that mediates the diverse biological effects of a multitude of ligands.

Minor comments

Page 8 line 20: ...each molecule.

Page 12 line 11: ... chicken to humans.

Page 13 line 10: ... we analyzed the NMR structure... It should say "... we analyzed the previously published NMR structure..." since the sentence seems to imply that the NMR structure was solved in this work.

Page 13 line 12: This revealed the Arg67, Ser 151 and Arg175 residues...: Arg175 is not shown in Fig. 6e.

Figure 1c: needs a better description in legend. It is confusing that the top blot line is labelled "pIR/IGF1R" since the blot shows all four receptors, and it is not clear what lines 1, 2 and 3 refer to (presumably triplicate experiments).

Signed: Pierre De Meyts

Reviewer #2 (Remarks to the Author)

The manuscript addresses the issue of the source of the different signalling outcomes of IGF-1R and IR, seeking to attribute them either to the extra-cellular or intra-cellular regions of the receptors (or both). Further, they show that in the intracellular IR residue Leu973 (IR-B), when

mutated to Phe (its IGF-1R equivalent), leads to an IGF-1R-like signalling outcome for the mutant insulin receptor. The latter finding is argued to arise from structural differences in Shc for peptides containing the Phe as opposed to the Leu.

The issue of the source of the differences in IR and IGF-1R signalling outcome has been investigated over many years. This is an important field (particular to those seeking to design therapeutics that target these receptors) and this study contributes remarkably to it. I find the manuscript very well written and the experiments carefully conducted with the data supportive of the conclusions. As such, I consider the manuscript highly suitable for publication in Nature Communications.

One topic that is, however not considered is the reason for mitogenic signalling by via IR-A upon IGF-II binding. While it may not be within the authors' capacity to undertake further experiments in this regard I consider that the issue should be addressed, as in the case of IGF-II the change in signalling outcome must be mediated via the extracellular rather than the intracellular domain.

The structural argument is not entirely convincing from Figures 6e and 6f. The existence of the pocket is reasonably clear in Fig 6e (as is its absence in Fig 6f) However, I notice the formation by the peptide of an additional strand to the beta sheet in Fig. 6f, so it seems to be possible that this might more than compensate the loss of the pocket for residue pY+1. I suggest that the argument be reworked and the figures improved (and also the PDB accession numbers from which they are based included in the text).

Finally, I don't believe that it is yet totally clear that IGF-1R has only one binding surface for IGF-1 (p19 of the ms), given that the second binding surface of insulin has been shown to have an homologous surface within IGF-1 [J Biol Chem (2008) 283, 20821-20829].

Reviewer #3 (Remarks to the Author)

Reviewer: Kenneth Siddle

This study reports a massive amount of very careful work and is by some distance the most thorough and wide-ranging on a topic of long-standing and widespread interest. It presents some fascinating data and significantly advances the field, most importantly by providing novel insights into structural features of IR and IGFR that may underlie their differential signalling capacity.

There are a number of relatively minor respects in which the manuscript might be improved.

1. Some additional experimental detail is required.

i) It should be clearly stated in Methods whether changes in gene expression after 6h insulin/IGF1 treatment are calculated relative to time 0 or 6h basal/untreated cells.

ii) It should be clearly stated whether there were any significant differences between clones in basal gene expression, especially in relation to differentially regulated genes.

ii) Fig 3b: It should be clearly stated how the plot was constructed, given that there are potentially 3 different clones for each construct. Do data points represent one representative construct or mean of all 3?

iii) Figs 4a and 5a: It should be clearly stated how the plots were constructed, given that there are two potential constructs for each ECD (Fig 4) and each ICD (Fig 5), and 3 different clones for each construct. Do data points represent mean of 6 constructs? Is it valid to pool data from two different constructs (wt and chimera)? (see also 2.iv below)

iv) Most of the gene expression data are presented as FCIR/FCIGF1R (or similar). Inspection of the the heat map in Figure 3 and scatter plot in Suppl Fig 3 suggests that for the overwhelming majority of genes the expression changes are <2-fold with both receptors. Many of the genes classed as differentially regulated according to an arbitrary 50% cut-off are in effect 'significantly' regulated by one receptor and not the other, while a few are regulated in the same sense by both receptors but to different extents. There is a further set of genes (including IRS-1 and Egrs) that are especially interesting as they appear to be regulated in opposite senses by IR and IGF1R. It would be useful to have these listed separately (if only as Supplementary data), and to see more detailed time-course data on their regulation.

2. Aspects of data presentation need further consideration.

i) It is a strength of the study that three independent clones have been obtained for each receptor construct (albeit the constructs are expressed at 10-20x the normal level of endogenous receptors). However, it seems that the studies of phosphorylation kinetics (Suppl Fig. 2), aimed primarily at identifying the most appropriate time point for subsequent studies, utilised only a single clone and there is no evidence of replication. The reproducibility of the patterns of phosphorylation is not convincingly established and is certainly not supported by statistical analysis, and description of some time courses as 'biphasic' is hard to justify (p.8, l.5-15).

ii) Patterns of substrate phosphorylation by different constructs (Fig 2) are quite complex and the description at times leans towards an over-simplified interpretation (eg p.8 l.21-26). The data for Shc are most convincing (Fig 2f), with phosphorylation by IGF1R and IR/IGF1R both much greater than IR and IGF1R/IR. However, this pattern is not so clearly mirrored in downstream components Erk1/2 (2d) and S6K1 (Fig. 2h), for both of which phosphorylation by IGF1R/IR is similar to IR/IGF1R. Importantly for IRS1 (Fig. 2b), although phosphorylation by IR/IGF1R is less than wild type IR, IGF1R is at least as great as IR, and there are no differences between constructs in terms of Akt phosphorylation (Fig. 2g) downstream of IRS-1. The text should comment on these anomalies. (Interestingly it is the Group I genes that, similarly to Shc-phosphorylation, show the most substantial IGF1R ICD-specific regulation, Fig. 5c).

iii) The text (p.14, l.11) claims there was decreased IRS-1 and Akt phosphorylation by IR L973F compared to wt IR, but in fact any difference is not statistically significant (Fig. 7 c/d) and the statement is potentially misleading (in contrast to the clearly increased phosphorylation of Shc, Gab, Erk, Fig 7 f/g/h).

iv) Some 597 individual genes are classed as specifically regulated by IR vs IGFR (p.9, l.14-16), 151 as specifically regulated by ECDs (p.10, l. 17) and 365 as specifically regulated by ICDs (p.11, l.10). Although a small number of individual genes are highlighted it would be useful to have some general representation (perhaps as a Venn diagram analogous to Fig. 3a) of where the overlaps lie in these comparisons. That would most robustly require separate comparisons of IR with IGFR/IR and of IGFR with IR/IGFR for ECDs, and of IR with IR/IGFR and of IGFR with IGFR/IR for ICDs (rather than pooled data for wt and relevant chimera as used for the comparisons at present).

3. Several points might be worthy of brief further discussion:

i) While it is unquestionably true to say that the biological processes regulated by IR and IGFR are "strikingly distinct at the physiological and pathological level" (p.4 l.16-17), this distinction surely owes much to differences in the levels of expression of receptors among tissues (IR being most highly expressed in terminally differentiated tissues such as liver, muscle and fat, while IGFR is most highly expressed in cell types undergoing proliferation). Studies of cells in vitro (including previous publications from this group, ref 18) have uniformly shown that IR and IGFR mediate similar metabolic and mitogenic effects (and regulate the expression of a similar spectrum of genes, as also shown in the present study) when examined in the same cell background, albeit with (relatively subtle) differences in the effectiveness with which they couple to different metabolic/mitogenic endpoints. Overall the evidence strongly implies that differences in signalling

specificity effectively 'fine tune' the different roles that are primarily a reflection of receptor distribution and ligand dynamics. This important point is worthy of mention in the Introduction and/or Discussion.

ii) The finding that ECDs may contribute to differential signalling is described as 'surprising' (p.19, l.16). However, the influence of ECDs, and of mechanisms and kinetics of ligand binding, on signalling outcomes has been highlighted and discussed previously and this should be acknowledged by appropriate citation (eg Jensen & De Meyts (2009) *Vitamin Horm* 80:51; Versteheye et al (2013) *Front Endocrinol* 4:98). More up to date references might also be cited in relation to the binding mechanisms of insulin and IGF-1 (eg Whitten et al (2009) *J Mol Biol* 394:878; Menting et al (2013) *Nature* 493:241; Menting et al (2015) *Structure* 23:1271).

iii) The data presented show a striking influence of L vs F at the +1 position relative to NPEY motif, in both IR and IGF1R (p.18, l.1-14). However published data relating to the role of this residue are complex. One previous study comparing the binding specificities of Shc and IRS-1 PTB domains (Wolf et al (1995) *J Biol Chem* 270:27407) showed that Shc PTB has a higher affinity for peptides from middleT, ErbB4 and EGFR (with S, W, L respectively at +1) than for the TrkA sequence (with F at +1), while IRS-1 PTB had highest affinity for the IL4R sequence (R at +1) than the IR sequence (L at +1). Another study (cited as ref 36 but not discussed in detail) found that although substitution of L at +1 with either A or R reduced Shc interaction by 70 and 90%, respectively, it had no effect upon interaction with IRS-1. Taken together these studies suggest that the +1 residue may have more influence on affinity for Shc than for IRS-1. As discussed above it is notable that IR and IGF1R (and wt and L973F IR) differ most markedly in terms of their capacity for Shc phosphorylation, and only modestly in terms of IRS-1 phosphorylation (see 2.ii/iii above). As Shc and IRS-1 compete for binding to the NPEY motif (cf Fig 6c), it may be that the +1 residue influences IRS binding indirectly, through its effect on competition with Shc, rather than by a direct effect on the affinity of IRS-1 per se. A corollary would be that differences in relative as well as the absolute expression levels of IRSs and Shc, as have been described in different cell types, may affect the balance between metabolic and mitogenic signalling (although of course both IRSs and Shc can engage Grb2 and thus mediate activation of the Ras/MAPK mitogenic pathway, and the IRS/PI3K pathway is also involved in mitogenic responses). This might merit a very brief discussion.

iv) Phosphorylation of IRS-1 and Shc as determined using site-specific antibodies (IRS-1 Y612 and Shc Y239/240) may not reflect overall phosphorylation of these substrates at the multiple sites known to be involved in recruiting downstream signalling molecules. Indeed there is published evidence that differential phosphorylation of IRS-1 by IR and IGF1R may be site-specific (Amoui et al (2001) *J Endocrinol* 171:153), and this work could be cited.

v) The authors comment in discussion p.19, l.4) on the Egr genes which, notwithstanding some differences in detail, have been shown to be rapidly (30 min) and transiently increased by a variety of mitogens, including both insulin and IGF1, in diverse cell types. The apparent differential down-regulation after 6 hrs treatment, as reported here, seems likely to be a late rebound effect following earlier induction of Egr1/2 by both IR and IGF1R, and it is difficult to interpret without a more detailed time course.

vi) Given the clear differences in substrate phosphorylation by IR and IGF1R, it is surprising that differential gene regulation is the exception rather than the rule, the large majority of genes being very similarly regulated by both receptors (as shown by several previous studies as well as this one). This paradox should be noted, perhaps with some brief speculation on the mechanisms by which a limited number of signalling pathways can deliver multiple differential responses (up or down regulation specific to IR or IGF1R) as well as (mostly) very similar responses to both receptors. This has a bearing on the concluding statement that the findings "will provide an opportunity for differentially targeting these pathways at a pharmacological level" (p.20, l.16).

Reviewer #1 (Remarks to the Author):

Insulin through binding to the insulin receptor (IR) and IGF-I and II binding to the IGF-I receptor (IGF1R) elicit distinct biological effects (primarily but not exclusively metabolic for insulin and primarily but not exclusively mitogenic for the IGFs). The structural basis of this signaling specificity has remained mysterious given the high degree of sequence homology between the two receptors, and the largely overlapping intracellular signaling pathways elicited by the receptor kinase activation. The problem is compounded by the existence of hybrid IR/IGF1R in cells that have both types of receptors. In this exciting paper, Cai, Sakaguchi et al. use an elegant approach to solve this conundrum. They generated brown preadipocytes in which both endogenous IR and IGF1R have been genetically inactivated using Cre-lox recombination (IR and IGF1R DKO cells). These DKO cells were then reconstituted with either wild-type mouse IR, IGF1R, or one of two chimeric receptors: IR/IGF1R with the IR extracellular domain (ECD) fused to the IGF1R transmembrane and intracellular domain (ICD) and IGF1R/IR with the ECD of IGF1R fused to the transmembrane and ICD of IR. These cell lines will have no hybrid receptors. The levels of receptor expression were similar in all cell lines. The ligand-dependent activation of the receptors was performed using the ligand matching the ECD of the overexpressed receptor.

All receptors showed a robust and similar level of receptor autophosphorylation. The mitogenic defect observed in the background DKO cells was rescued by the wild-type IGF1R, but less well by the IR. The constructs containing the IGF1R ICD showed a greater mitogenic potential than those with IR ICD. The opposite was true for metabolic activity. Receptors with IR ECD showed more rapid internalization. There were also marked differences in downstream targets phosphorylation by the four receptors. The intracellular domain of IR was more potent in phosphorylating IRS-1 while the ICD of IGF1R was better in activating Shc, Gab-1 and p70S6K1 signaling pathways. The authors also studied the gene expression patterns elicited by the four receptors and showed marked differences depending on both the ECDs and ICDs, with the IR domains being more associated with metabolic signaling. Finally, the authors examined the role of the juxtamembrane domain NPEY motif and showed that it confers the observed specificity of the ICD, and by site-directed mutagenesis showed that substituting Leu 973 to Phe (like in IGF1R) in the IR ICD shifts the signaling pattern to that of the IGF-1R, showing that this residue plays a critical role in substrate preference. The authors propose a structural interpretation based on the NMR structure of the Shc PTB domain bound to a phosphorylated NPEY domain of the TrkA receptor. The overall major conclusion is that the differences in insulin and IGF-I action are due to differences in the juxtamembrane ICDs of their receptors, especially in sequence difference at position 973 in the IR and 951 in the IGF1R. These intrinsic receptor differences contribute to differences in substrate phosphorylation between these ligands and in their ability to exert metabolic control versus mitogenic regulation.

The article is convincing and well written, and nicely illustrated. The experimental approach is logical and straightforward, and complex experiments are presented in an accessible way. The work brings a novel and important element for discussion of the molecular and structural basis of intracellular signaling specificity.

One has to remain aware though that this elegant reductionist approach markedly simplifies the system by having cell lines with only one receptor or receptor chimera stimulated by one ligand, while in real life most cells will have both IR and IGF1R plus hybrids, being stimulated by three ligands, insulin, IGF-I and IGF-II. The combinatorial nature of signaling under such complex conditions remains to be explored.

We thank the reviewer for bringing up this important point. We acknowledge that in normal cells and in vivo, other factors, like the relative expression of IR, IGF1R and hybrid receptors and cross activation by different ligands, contribute to the different signaling and functions controlled by IR and IGF1R. Indeed, it was because of this complexity of the IR and IGF1R system that we designed the current cell models in

which IR and IGF1R null cells were used to express only one receptor type at a time, thereby eliminating the potential formation of the hybrid receptors and ligand cross-activation. With this clean system, we identified a previously unknown mechanism, by which IR and IGF1R differ their substrate preference and regulation of gene expression. This new mechanism expands our understanding of the differential signaling by IR and IGF1R. And we will continue to explore the physiological significance in the future.

I have only one substantial critique and some very minor ones.

A weak part of the paper is the discussion of the role of the extracellular domain of IR and IGF1R in differential signaling. The authors state the following:

"The IR extracellular domain harbors two interactions domains, one with high affinity and the other one with low affinity. binding for insulin, providing unique binding kinetics for negative cooperativity. By contrast, the IGF1R has only one IGF-I binding surface consisting of the cysteine-rich region and the flanking regions from L1 and L2 domains".

The statement regarding the IGF1R having only one binding surface is incorrect and ignores most of the recent advances in the structural biology of IGF1R binding. Firstly, the IGF1R also exhibits negative cooperativity in binding, suggesting a similar binding mode to that of insulin (Christoffersen CT et al., *Endocrinology* 135:4772-475, 1994). One difference is that the IGF1R dose-response curve is not bell-shaped. Secondly, a second binding site homologous to insulin's binding site 2 has been identified on IGF-I by alanine scanning mutagenesis, made of Glu9, Asp12, Phe16, Leu54 and Glu 58. Multiple substitutions in this site 2 result in a 33 to 100-fold reduction in affinity for the IGF1R (Gauguin L et al., *J Biol Chem* 283:20821-20829, 2008). Likewise, site 2 on IGF-II has been mapped (Albino CL et al., *J Biol Chem* 284:7656-7664, 2009). Thirdly, mathematical modeling of insulin and IGF1R binding kinetics has quantitatively accounted for all the kinetic and equilibrium binding properties of both IR and IGF1R based on the same bivalent crosslinking model (Kiselyov VV et al., *Mol Syst Biol* 5:243, 2009). The authors are correct in that for the IGF1R site 1 extends into the cysteine-rich region which has been previously shown to accommodate the C-domain absent in insulin.

We appreciate the comments and suggested correction by the reviewer. We have included more thorough discussion regarding ligand binding in the Discussion. (p.19, l.6-16)

One point not discussed by the authors which may deserve mention is that the ligands themselves contain structural information that has some role in determining differential signaling through the same receptor. It has been shown in cells only expressing IGF1R that insulin, IGF-I and IGF-II elicit overlapping but non-identical patterns of gene expression, with some genes regulated by only one of the ligands (*Front Endocrinol (Lausanne)* 4:98, 2013). This is biologically highly relevant since unlike Vertebrates, Invertebrates have only one insulin-like receptor that mediates the diverse biological effects of a multitude of ligands.

This is another great point by the reviewer. The different responses of different ligand even on the same receptor have been reported and cannot be ignored. Indeed, our cell model, in which only one type of receptors is expressed, is a great system to further explore this question. However, with all of the other data in the paper, we felt that adding such experiments to the present study would be impossible. We have, however, incorporated a discussion of this point in the revised manuscript. (p.19, l.16-18) and plan to do such experiments in future studies.

Minor comments

Page 8 line 20: ...each molecule.

Page 12 line 11: ... chicken to humans.

Page 13 line 10: ... we analyzed the NMR structure... It should say "... we analyzed the previously published NMR structure..." since the sentence seems to imply that the NMR structure was solved in this work.

We have made the above-suggested changes in the revised manuscript.

Page 13 line 12: This revealed the Arg67, Ser 151 and Arg175 residues...: Arg175 is not shown in Fig. 6e.

Thank you for the comment. In the structural domain shown we wanted to focus on the Y+1 position and therefore, it is hard to include Arg175 in the figure. In the revised version, we have modified this statement to focus on the Y+1 position and SHC binding and have deleted the description of the phosphotyrosine-coordinating residues as this is no longer necessary.

Figure 1c: needs a better description in legend. It is confusing that the top blot line is labelled "pIR/IGF1R" since the blot shows all four receptors, and it is not clear what lines 1, 2 and 3 refer to (presumably triplicate experiments).

We apologize for the confusing labeling. The phospho-specific antibody used in these experiments and by many workers in the field (Cell signaling, Catalog# 3024) recognizes the phosphorylation of sequential tyrosine residues in the kinase domains of both IR and IGF1R, since the sequence of the receptors is the same at the site. Therefore, all the cell lines would have signals in response to ligand stimulation, indicating the activation of the receptors. We have changed the labeling in the figures from "pIR/IGF1R" to "pIR/pIGF1R" and better described the specificity of the antibody in the Methods.

Signed: Pierre De Meyts

Reviewer #2 (Remarks to the Author):

The manuscript addresses the issue of the source of the different signaling outcomes of IGF-1R and IR, seeking to attribute them either to the extra-cellular or intra-cellular regions of the receptors (or both). Further, they show that in the intracellular IR residue Leu973 (IR-B), when mutated to Phe (its IGF-1R equivalent), leads to an IGF-1R-like signaling outcome for the mutant insulin receptor. The latter finding is argued to arise from structural differences in Shc for peptides containing the Phe as opposed to the Leu.

The issue of the source of the differences in IR and IGF-1R signaling outcome has been investigated over many years. This is an important field (particular to those seeking to design therapeutics that target these receptors) and this study contributes remarkably to it. I find the manuscript very well written and the experiments carefully conducted with the data supportive of the conclusions. As such, I consider the manuscript highly suitable for publication in Nature Communications.

One topic that is, however not considered is the reason for mitogenic signalling by via IR-A upon IGF-II binding. While it may not be within the authors' capacity to undertake further experiments in this regard I consider that the issue should be addressed, as in the case of IGF-II the change in signalling outcome must be mediated via the extracellular rather than the intracellular domain.

We appreciate the comment by the reviewer, which is similar to one of the points raised by Reviewer 1, namely that different ligands occupying the same receptor may have different responses. This could be

due to differences in conformational change following ligand binding to the extracellular domain of the receptor or how this change is propagated to the intracellular domains of the receptor. However, a detailed study of this question is not within the scope of the present study. As suggested, however, we have incorporated some discussion of this point in the revised manuscript highlighting the possibility of different ligand binding could contribute to different signaling. (p.19, l.16-18)

The structural argument is not entirely convincing from Figures 6e and 6f. The existence of the pocket is reasonably clear in Fig 6e (as is its absence in Fig 6f) However, I notice the formation by the peptide of an additional strand to the beta sheet in Fig. 6f, so it seems to be possible that this might more than compensate the loss of the pocket for residue pY+1. I suggest that the argument be reworked and the figures improved (and also the PDB accession numbers from which they are based included in the text).

Thanks for the comment. Actually, the beta-strand interaction is formed in both the SHC and IRS-1 structures. We have revised the text to make this clear. (Note that PyMOL, the program used to prepare this figure, does not represent this segment in the SHC/TrkA structure as a beta-sheet because the phi/psi angles are somewhat outside its default ranges, and we were reluctant to over-ride the defaults.)

We have also added figure panels using a surface representation for the PTB domains (as Supp. Fig 7) to better show the binding pockets. PDB IDs are now given in the Figure legend and the corresponding papers cited in the text.

Finally, I don't believe that it is yet totally clear that IGF-1R has only one binding surface for IGF-1 (p19 of the ms), given that the second binding surface of insulin has been shown to have an homologous surface within IGF-1 [J Biol Chem (2008) 283, 20821-20829].

Thank you for the comment. As noted in response to Reviewer 1, we have now included a more thorough discussion regarding the ligand binding mechanisms in the Discussion section of the paper. (p.19, l.6-16)

Reviewer #3 (Remarks to the Author):

Reviewer: Kenneth Siddle

This study reports a massive amount of very careful work and is by some distance the most thorough and wide-ranging on a topic of long-standing and widespread interest. It presents some fascinating data and significantly advances the field, most importantly by providing novel insights into structural features of IR and IGFR that may underlie their differential signalling capacity.

There are a number of relatively minor respects in which the manuscript might be improved.

1. Some additional experimental detail is required.

i) It should be clearly stated in Methods whether changes in gene expression after 6h insulin/IGF1 treatment are calculated relative to time 0 or 6h basal/untreated cells.

We appreciate the suggestions by the reviewer. The changes of gene expression are calculated by comparing 6 h stimulated samples with 6 h mock treated samples. We have made edits in the Materials and Methods section to make this point clear. (p.23, l.13-14)

ii) It should be clearly stated whether there were any significant differences between clones in basal gene expression, especially in relation to differentially regulated genes.

We have now included a new PCA plot as Supplementary Figure 3 in which we compare the overall gene expression profiles among all the cells types in the basal and stimulated states. The 4 cell types showed slightly different overall gene expression profiles even in the basal state and serum starved, indicating expression of different types of receptors can lead to different patterns of gene expression even in the basal state. More importantly, ligand stimulation induced dramatic hormone-dependent effects (p.9, 1.2-3). In this manuscript, we focused on the ligand response, which is the fold changes of expression of each gene in response to stimulation in all the cell types, but have now added a comment about the differences in basal gene expression and a reference to the new Supplementary Figure.

ii) Fig 3b: It should be clearly stated how the plot was constructed, given that there are potentially 3 different clones for each construct. Do data points represent one representative construct or mean of all 3?

Each data point represents the average value of 3 different clones of each receptor construct in all the volcano plots and scattered plots in the manuscript. One clone of cells expressing the IR/IGF1R construct (clone 2) was an outlier as judged by PCA analysis and received a low quality weight. This was therefore excluded in the subsequent analysis. These points have now been clarified in the figure legends in the revised manuscript.

iii) Figs 4a and 5a: It should be clearly stated how the plots were constructed, given that there are two potential constructs for each ECD (Fig 4) and each ICD (Fig 5), and 3 different clones for each construct. Do data points represent mean of 6 constructs? Is it valid to pool data from two different constructs (wt and chimera)? (see also 2.iv below)

We appreciate the comment by the reviewer. In the scattered plots in Figures 4 and 5, each data point represents the mean value of all the constructs containing the same ECD or the same ICD. Specifically, IR-ICD (3 clones of IR, 3 clones of IGF1R/IR), IGF1R-ICD (3 clones of IGF1R, 2 clones of IR/IGF1R), IR-ECD (3 clones of IR, 2 clones of IR/IGF1R) and IGF1R-ECD (3 clones of IGF1R, 3 clones of IGF1R/IR). As noted above, one clone of IR/IGF1R was an outlier and excluded from the bioinformatics analysis. We recognize both ICD and ECD have effects on gene expression regulation. To simplify the model and focus on the common ICD or ECD-dependent differences between IR and IGF1R, we pooled data from wild-type receptor with corresponding chimera with the same ICD or ECD of the receptors. In this way, we were able to focus on the ECD-dependent differences of IR and IGF1R in Fig 4A. and ICD-dependent differences of IR and IGF1R in Fig 5A. The top candidate genes were confirmed by qPCR, validating our bioinformatics analysis.

iv) Most of the gene expression data are presented as FCIR/FCIGF1R (or similar). Inspection of the the heat map in Figure 3 and scatter plot in Suppl Fig 3 suggests that for the overwhelming majority of genes the expression changes are <2-fold with both receptors. Many of the genes classed as differentially regulated according to an arbitrary 50% cut-off are in effect 'significantly' regulated by one receptor and not the other, while a few are regulated in the same sense by both receptors but to different extents.

There is a further set of genes (including IRS-1 and Egrs) that are especially interesting as they appear to be regulated in opposite senses by IR and IGF1R. It would be useful to have these listed separately (if only as Supplementary data), and to see more detailed time-course data on their regulation.

From the microarray data, IRS-1 and Egr1/2 appears to have the opposite regulation by IR and IGF1R, however, qPCR confirmation showed Egr1/2 is highly suppressed in IR and IR-

ICD cells, but not regulated in IGF1R and IGF1R-ICD cells. Similarly, as shown below, IRS-1 expression is more suppressed by IR and IGF1R/IR (IR-ICD), but not significantly regulated by IGF1R, and IR/IGF1R (IGF1R-ICD). Therefore, we choose not to list these genes separately.

In addition, we agree that the early response genes like, Egr1 and Egr2, are regulated by many mitogens, with increases at early time points and decreases at later time point. Therefore, it would be more appropriate to assess the expression of these two genes at early time point (30 min) and later ones (6 h). We included all these data into a new Supplementary Figure 6. Indeed, at the early time point (30 min), Egr1 and Egr2 expressions were more extensively induced by the receptors with IGF1R-ICD and at the later time point (6 h), more potently suppressed by the receptors with IR-ICD. (p.11, l.17-22)

2. Aspects of data presentation need further consideration.

i) It is a strength of the study that three independent clones have been obtained for each receptor construct (albeit the constructs are expressed at 10-20x the normal level of endogenous receptors). However, it seems that the studies of phosphorylation kinetics (Suppl Fig. 2), aimed primarily at identifying the most appropriate time point for subsequent studies, utilised only a single clone and there is no evidence of replication. The reproducibility of the patterns of phosphorylation is not convincingly established and is certainly not supported by statistical analysis, and description of some time courses as ‘biphasic’ is hard to justify (p.8, l.5-15).

We agree that it would be more appropriate to analyze all the three clones of each cell type for both signaling kinetics and amplitudes. Therefore, we have now repeated the time course of signaling in all the other clones of the cells and represented the data as mean + SEM in revised Supplementary Figure 2. We have also made the necessary changes in the main text of the manuscript to describe the new figure. (p.7, l.24 – p.8, l.6)

ii) Patterns of substrate phosphorylation by different constructs (Fig 2) are quite complex and the description at times leans towards an over-simplified interpretation (eg p.8 l.21-26). The data for Shc are most convincing (Fig 2f), with phosphorylation by IGF1R and IR/IGF1R both much greater than IR and IGF1R/IR. However, this pattern is not so clearly mirrored in downstream components Erk1/2 (2d) and S6K1 (Fig. 2h), for both of which phosphorylation by IGF1R/IR is similar to IR/IGF1R. Importantly for IRS1 (Fig. 2b), although phosphorylation by IR/IGF1R is less than wild type IR, IGF1R is at least as great as IR, and there are no differences between constructs in terms of Akt phosphorylation (Fig. 2g) downstream of IRS-1. The text should comment on these anomalies. (Interestingly it is the Group I genes that, similarly to Shc-phosphorylation, show the most substantial IGF1R ICD-specific regulation, Fig. 5c).

Thank you for this important comment. We have made the suggested changes in both the results and discussion sections and highlighted in the revised manuscript. (p.8, l.13-14; p.18, l.1-7; p.18, l.20-23)

iii) The text (p.14, l.11) claims there was decreased IRS-1 and Akt phosphorylation by IR L973F compared to wt IR, but in fact any difference is not statistically significant (Fig. 7 c/d) and the statement is potentially misleading (in contrast to the clearly increased phosphorylation of Shc, Gab, Erk, Fig 7 f/g/h).

We agree with the Reviewer’s point. The data show that both IRS-1 and Akt phosphorylation exhibit a **trend toward a decrease** in cells expressing IR L973F compared with those in cells expressing normal IR, but it is not statistically significant. We have made the necessary changes in the revised manuscript. (p.14, l.17-18; p.17, l.14)

iv) Some 597 individual genes are classed as specifically regulated by IR vs IGFR (p.9, 1.14-16), 151 as specifically regulated by ECDs (p.10, 1. 17) and 365 as specifically regulated by ICDs (p.11, 1.10). Although a small number of individual genes are highlighted it would be useful to have some general representation (perhaps as a Venn diagram analogous to Fig. 3a) of where the overlaps lie in these comparisons. That would most robustly require separate comparisons of IR with IGFR/IR and of IGFR with IR/IGFR for ECDs, and of IR with IR/IGFR and of IGFR with IGFR/IR for ICDs (rather than pooled data for wt and relevant chimera as used for the comparisons at present).

We have included a new Supplementary Figure 5, representing the numbers of the genes that are regulated corresponding to the ICD and ECD of the receptors in either IR or IGF1R-regulated gene pools. The new data show that ICD of the receptors have a much stronger effect on gene expression, whereas the ECD has some, but limited, effects on regulation of gene expression. Interestingly, there is also a significant portion of the genes that are uniquely regulated by either IR or IGF1R that do not appear to be regulated by either the ICD or ECD alone of the receptors, suggesting for these events both domains are required for the appropriate gene regulation. It is also possible that with $n = 3$, we simply don't have enough statistical power to identify all the genes corresponding to the ICD and ECD of the receptors. Unfortunately, for practical expense reasons we could not power the study with a larger number of clones of each receptor subtype.

3. Several points might be worthy of brief further discussion:

i) While it is unquestionably true to say that the biological processes regulated by IR and IGFR are “strikingly distinct at the physiological and pathological level” (p.4 1.16-17), this distinction surely owes much to differences in the levels of expression of receptors among tissues (IR being most highly expressed in terminally differentiated tissues such as liver, muscle and fat, while IGFR is most highly expressed in cell types undergoing proliferation). Studies of cells in vitro (including previous publications from this group, ref 18) have uniformly shown that IR and IGFR mediate similar metabolic and mitogenic effects (and regulate the expression of a similar spectrum of genes, as also shown in the present study) when examined in the same cell background, albeit with (relatively subtle) differences in the effectiveness with which they couple to different metabolic/mitogenic endpoints. Overall the evidence strongly implies that differences in signalling specificity effectively ‘fine tune’ the different roles that are primarily a reflection of receptor distribution and ligand dynamics. This important point is worthy of mention in the Introduction and/or Discussion.

Again, we appreciate this point. The differential roles of IR and IGF1R have many contributing factors. The distinct expression patterns of these two receptors certainly contributes to the differential functions of the IR and IGF1R in the whole body. We have already included a section in the introduction to acknowledge this (p.4, 1.26 – p.5, 1.2). Our present study does not ignore or devalue this contributing factor. However, in the present study, we are focusing on the contributing factors derived from receptor themselves and the domain-dependent effects of IR and IGF1R.

ii) The finding that ECDs may contribute to differential signalling is described as ‘surprising’ (p.19, 1.16). However, the influence of ECDs, and of mechanisms and kinetics of ligand binding, on signalling outcomes has been highlighted and discussed previously and this should be acknowledged by appropriate citation (eg Jensen & De Meyts (2009) Vitamin Horm 80:51; Verstehey et al (2013) Front Endocrinol 4:98). More up to date references might also be cited in relation to the binding mechanisms of insulin and IGF-1 (eg Whitten et al (2009) J Mol Biol 394:878; Menting et al (2013) Nature 493:241; Menting et al (2015) Structure 23:1271).

We have included more discussion and references in the revised manuscript as suggested. (p.19, 1.4-18)

iii) The data presented show a striking influence of L vs F at the +1 position relative to NPEY motif, in

both IR and IGF1R (p.18, 1.1-14). However published data relating to the role of this residue are complex. One previous study comparing the binding specificities of Shc and IRS-1 PTB domains (Wolf et al (1995) J Biol Chem 270:27407) showed that Shc PTB has a higher affinity for peptides from middleT, ErbB4 and EGFR (with S, W, L respectively at +1) than for the TrkA sequence (with F at +1), while IRS-1 PTB had highest affinity for the IL4R sequence (R at +1) than the IR sequence (L at +1). Another study (cited as ref 36 but not discussed in detail) found that although substitution of L at +1 with either A or R reduced Shc interaction by 70 and 90%, respectively, it had no effect upon interaction with IRS-1. Taken together these studies suggest that the +1 residue may have more influence on affinity for Shc than for IRS-1.

As discussed above it is notable that IR and IGF1R (and wt and L973F IR) differ most markedly in terms of their capacity for Shc phosphorylation, and only modestly in terms of IRS-1 phosphorylation (see 2.ii/iii above). As Shc and IRS-1 compete for binding to the NPEY motif (cf Fig 6c), it may be that the +1 residue influences IRS binding indirectly, through its effect on competition with Shc, rather than by a direct effect on the affinity of IRS-1 per se. A corollary would be that differences in relative as well as the absolute expression levels of IRSs and Shc, as have been described in different cell types, may affect the balance between metabolic and mitogenic signalling (although of course both IRSs and Shc can engage Grb2 and thus mediate activation of the Ras/MAPK mitogenic pathway, and the IRS/PI3K pathway is also involved in mitogenic responses). This might merit a very brief discussion.

We appreciate and agree with the reviewer's comment. The mutagenesis studies, including several listed by the reviewer, showed that the NPEY motif in IR uses different residues for the substrate recognition. Briefly, Leu at Y-8, Tyr at Y-7, Asn at Y-3, Pro at Y-2 and Ala at Y+3 positions are all involved in IRS-1 binding, whereas Ser at Y-4, Asn at Y-3, Pro at Y-2 and Leu at Y+1 positions are important for Shc binding. Among these residues, only Y+1 residue is not conserved between IR and IGF1R, thus providing a candidate residue for the differential binding affinities of Shc for the IR-NPEY and IGF1R-NPEY domains. Indeed, our Co-IP studies clearly showed that receptors with NPEY motif with Phe at the Y+1 position have much higher binding affinity for Shc than those with NPEY motifs with Leu at Y+1 position. Furthermore, the Shc/IRS-1 competition binding assay for the receptor (Fig. 6c) showed that Shc bound to IR L973 better and was much more efficient in competing against IRS-1 for IR L973 binding. Together, the Y+1 residue directly affects Shc binding, therefore indirectly affecting IRS-1 recruitment. We have made changes in the manuscript to emphasize this point. (p.13, 1.7-8; p.14, 1.4-6)

iv) Phosphorylation of IRS-1 and Shc as determined using site-specific antibodies (IRS-1 Y612 and Shc Y239/240) may not reflect overall phosphorylation of these substrates at the multiple sites known to be involved in recruiting downstream signalling molecules. Indeed there is published evidence that differential phosphorylation of IRS-1 by IR and IGF1R may be site-specific (Amoui et al (2001) J Endocrinol 171:153), and this work could be cited.

In the present manuscript, we only assessed the most common phosphorylation sites of each signaling molecules using commonly used phospho-specific antibodies. Our data show clearly different phosphorylation patterns of key signaling molecules in cells expressing different receptor types, especially the phosphorylation of Shc and IRS-1, which strongly corresponding to the different gene expression pattern in the same group of cells. However, we also acknowledge that a more comprehensive method could be used to more fully assess the phosphorylation of intracellular proteins in these cells. The potential phosphoproteomic study is proposed and discussed in the revised manuscript. (p.18, 1.1-7)

v) The authors comment in discussion p.19, 1.4) on the Egr genes which, notwithstanding some differences in detail, have been shown to be rapidly (30 min) and transiently increased by a variety of mitogens, including both insulin and IGF1, in diverse cell types. The apparent differential down-

regulation after 6 hrs treatment, as reported here, seems likely to be a late rebound effect following earlier induction of Egr1/2 by both IR and IGF1R, and it is difficult to interpret without a more detailed time course.

Please see our response for point 1.vi above. Briefly, we analyzed Egr1 and 2 expression levels at 30 min and 6 h after stimulation, and showed that these two early response genes are differentially regulated in a dynamic fashion corresponding to the intracellular domains of the IR and IGF1R. (p.11, l.17-22)

vi) Given the clear differences in substrate phosphorylation by IR and IGF1R, it is surprising that differential gene regulation is the exception rather than the rule, the large majority of genes being very similarly regulated by both receptors (as shown by several previous studies as well as this one). This paradox should be noted, perhaps with some brief speculation on the mechanisms by which a limited number of signalling pathways can deliver multiple differential responses (up or down regulation specific to IR or IGF1R) as well as (mostly) very similar responses to both receptors. This has a bearing on the concluding statement that the findings “will provide an opportunity for differentially targeting these pathways at a pharmacological level” (p.20, l.16).

In the present study, as well as other previous studies, regulation of gene expression by IR and IGF1R is largely overlapping with only a relatively small portion of genes differentially regulated by either IR or IGF1R. This is not surprising because both IR and IGF1R activate several common signaling pathways. However, as the current study shows, the amplitude of response for a select number of downstream signaling cascades differs. Accordingly, most of the genes are regulated similarly in IR and IGF1R, and a smaller portion of the genes showed more potent regulation in one receptor over the other. Our data show that Shc signaling likely contributes to a significant portion of IGF1R-ICD dependent gene expression. However, the detailed molecular mechanism still remains elusive and requires further investigation. We have incorporated a brief discussion commenting on this point. (p.18, l.20-23)

Reviewer #1 (Remarks to the Author)

The authors have satisfactorily addressed all the critiques of the reviewers, and this excellent paper is now suitable for publication.

Pierre De Meyts

Reviewer #2 (Remarks to the Author)

The revisions to the manuscript are adequate to meet my original comments / criticisms as Reviewer #2. From my point of view the manuscript is now suitable for publication.

Reviewer #3 (Remarks to the Author)

The authors have provided a commendably detailed response to all the reviewers, and have made several significant revisions to the manuscript, which both clarify the data and expand the discussion.